# Encoding and control of orientation to airflow by a set of *Drosophila* fan-shaped body neurons

**Timothy A Currier[1,2], Andrew MM Matheson[1], Katherine I Nagel[1,2]***

[1]Neuroscience Institute, New York University Langone Medical Center, New York, United States; [2]Center for Neural Science, New York University, New York, United States

**Abstract** The insect central complex (CX) is thought to underlie goal-oriented navigation but its functional organization is not fully understood. We recorded from genetically-identified CX cell types in *Drosophila* and presented directional visual, olfactory, and airflow cues known to elicit orienting behavior. We found that a group of neurons targeting the ventral fan-shaped body (ventral P-FNs) are robustly tuned for airflow direction. Ventral P-FNs did not generate a 'map' of airflow direction. Instead, cells in each hemisphere were tuned to 45° ipsilateral, forming a pair of orthogonal bases. Imaging experiments suggest that ventral P-FNs inherit their airflow tuning from neurons that provide input from the lateral accessory lobe (LAL) to the noduli (NO). Silencing ventral P-FNs prevented flies from selecting appropriate corrective turns following changes in airflow direction. Our results identify a group of CX neurons that robustly encode airflow direction and are required for proper orientation to this stimulus.

**\*For correspondence:**
katherine.nagel@nyulangone.org

**Competing interests:** The authors declare that no competing interests exist.

## Introduction

Foraging for food, locating mates, and avoiding predation all depend on an animal's ability to navigate through complex multi-sensory environments. Many animals are known to compare and combine visual, mechanosensory and olfactory cues to achieve their navigational goals (*Gire et al., 2016*; *Holland et al., 2009*; *Bianco and Engert, 2015*; *Dacke et al., 2019*; *Cardé and Willis, 2008*; *Lockery, 2011*). Identifying the brain regions and circuit organizations that support navigation with respect to different modalities is a fundamental question in neuroscience.

In insects, a conserved brain region known as the central complex (CX) is thought to control many aspects of navigation (*Strauss and Heisenberg, 1993*; *Honkanen et al., 2019*). The CX is a highly organized neuropil consisting of four primary subregions: the protocerebral bridge (PB), the ellipsoid body (EB), the fan-shaped body (FB), and the paired noduli (NO). Columnar neurons recurrently connect these regions to each other, while tangential neurons targeting different layers of the EB and FB provide a large number of inputs from the rest of the brain (*Hanesch et al., 1989*; *Wolff et al., 2015*; *Franconville et al., 2018*; *Hulse et al., 2020*). Outputs are provided by different subsets of columnar neurons (*Stone et al., 2017*; *Franconville et al., 2018*; *Scheffer et al., 2020*; *Hulse et al., 2020*).

Recent work has led to a burgeoning understanding of how the CX is functionally organized. In the EB, a group of 'compass neurons' (or E-PGs) exhibit an abstract map of heading angle that is derived from both visual and airflow landmark cues (*Seelig and Jayaraman, 2015*; *Green et al., 2017*; *Fisher et al., 2019*; *Shiozaki et al., 2020*; *Okubo et al., 2020*). Another set of EB neurons, known as P-ENs, rotate this heading representation when the fly turns in darkness (*Green et al., 2017*; *Turner-Evans et al., 2017*). Despite these robust representations of navigation-relevant variables, the EB compass network is not required for all forms of goal-directed navigation. Silencing

E-PGs disrupts menotaxis—straight-line navigation by keeping a visual landmark at an arbitrary angle—but not other kinds of visual orienting (*Giraldo et al., 2018*; *Green et al., 2019*).

In contrast, the FB may influence ongoing locomotor activity more directly. For example, cockroaches alter their climbing and turning strategies when the FB is lesioned (*Harley and Ritzmann, 2010*), while FB stimulation evokes stereotypic walking maneuvers (*Martin et al., 2015*). However, 'compass'-like signals encoding heading and steering are also present in some parts of the FB (*Shiozaki et al., 2020*). Columnar neurons of the FB have been proposed to represent a desired heading, while output neurons of the FB have been proposed to drive steering (*Stone et al., 2017*; *Honkanen et al., 2019*), but these hypotheses have not been directly tested experimentally. How the FB participates in navigation, and whether its role is distinct from that of the EB, is currently unclear.

As in the EB, FB neurons represent a wide array of sensory cues, including optic flow, polarized light, and mechanical activation of the antennae or halteres (*Weir and Dickinson, 2015*; *Heinze et al., 2009*; *Ritzmann et al., 2008*; *Phillips-Portillo, 2012*; *Kathman and Fox, 2019*). Although it has received less attention than vision or olfaction, flow of the air or water is a critical mechanosensory cue for animals navigating in aquatic, terrestrial, and air-borne environments (*Montgomery et al., 1997*; *Yu et al., 2016a*; *Alerstam et al., 2011*; *Reynolds et al., 2010*). The primary sensors that detect flow are well-described in many species (*Suli et al., 2012*; *Yu et al., 2016b*; *Yorozu et al., 2009*), but an understanding of the higher brain circuits that process flow signals is just beginning to emerge (*Okubo et al., 2020*; *Suver et al., 2019*). The neurons and computations that directly support flow-based navigation remain unknown.

Here, we used whole-cell recordings to systematically investigate the sensory responses of many of the major columnar cell types in the CX in open loop. We measured responses to three stimuli known to elicit basic orienting responses in *Drosophila*: a visual stripe, directional airflow, and an attractive odor. We found that columnar neurons targeting the ventral layers of the FB and the third compartment of the nodulus ('ventral P-FNs') were robustly tuned for the direction of airflow, but not our other stimuli. The ventral P-FN category contains the PFNa, PFNm, and PFNp cell types identified in recent connectomics studies (*Scheffer et al., 2020*; *Hulse et al., 2020*). Recordings from different columns suggest that ventral P-FN sensory responses are not organized in a sensory 'compass' or 'map'—where all possible stimulus directions are represented (*Fisher et al., 2019*; *Okubo et al., 2020*). Instead, ventral P-FNs primarily encode airflow arriving from two directions, approximately 45° to the right and left of the midline. Single neuron tuning depended on the hemisphere in which its cell body was located, with each FB column innervated by both left- and right-preferring neurons. Imaging and recording experiments suggest that this airflow representation may be inherited from the lateral accessory lobe (LAL), which projects to the third nodulus compartment ($NO_3$) in each hemisphere. This anatomy could explain why all ventral P-FNs in one hemisphere share the same sensory tuning.

Genetic silencing experiments suggest that ventral P-FNs are required for normal orientation to airflow in a closed-loop flight simulator. Flies with silenced ventral P-FNs fail to make appropriate corrective turns in response to a change in airflow direction, but respond normally to airflow pauses, arguing for a specific role in linking directional sensory input to corrective motor actions. Our results support the hypothesis that different CX compartments represent sensory information in distinct formats, and identify a neural locus in the ventral FB that promotes orientation to airflow.

## Results

### Airflow dominates responses to directional sensory cues in a set of CX columnar neurons

To assess how CX compartments might differentially process sensory cues to guide navigation, we first surveyed columnar cell types that target the PB and different layers of the EB and FB. Our survey included: (1) all known columnar cell types that link the PB and NO ('P-XN' neurons); and (2) two additional cell types that target regions outside the CX proper, instead of the NO. Many of these cell types have not been previously recorded using electrophysiology. We used publicly available split-GAL4 lines (*Wolff and Rubin, 2018*) to express GFP in each population, then made whole-cell recordings while we presented flies with the following sensory cues, either alone or in combination:

a high contrast vertical stripe, airflow generated by a pair of tubes, and apple cider vinegar, which could be injected into the airstream (*Figure 1A and B*, right). Flies fixate vertical stripes while walking and in flight (*Reichardt and Poggio, 1976*; *Heisenberg and Wolf, 1979*; *Maimon et al., 2008*) and tend to orient away from an airflow source (*Currier and Nagel, 2018*; *Kaushik et al., 2020*). The addition of an attractive odorant to airflow switches orientation from downwind to upwind (*van Breugel and Dickinson, 2014*; *Álvarez-Salvado et al., 2018*). We presented each stimulus combination from four directions: frontal, rear, ipsilateral, and contralateral (*Figure 1B*, left). We monitored fly activity with an infrared camera and discarded the few trials that contained flight behavior.

We first noticed that CX columnar cell types possessed diverse baseline activity (*Figure 1—figure supplement 1* and *Table 1*). Some cell types, such as P-EN2 and P-F$_2$N$_3$, showed rhythmic fluctuations in membrane potential, evident in the timecourses and distributions of membrane potential (*Figure 1C*). Rhythmic neurons exhibited broader membrane potential distributions than less rhythmic neurons (*Figure 1C*, right and *Figure 1—figure supplement 1B&C*). P-F$_3$LC, for example, showed stable baseline activity and a lower characteristic resting potential. We did not observe any correlation between fly behavior and the presence or absence of oscillations, but resting membrane potential did rarely fluctuate with leg movements. Input resistance varied by cell type, with values ranging from 1.5 to 10 GOhm (*Figure 1—figure supplement 1D*).

We next turned our attention to the sensory responses of CX columnar cell types. We found that, although most neurons responded to each cue in some manner, airflow responses were generally larger than stripe responses across cell types in the survey. In P-F$_1$N$_3$ neurons, for example, large directionally tuned spiking responses were observed during airflow presentation, but not stripe presentation (*Figure 1D*). To assess this difference across cell types, we identified the cue direction(s) that elicited the largest stripe and airflow responses for each recorded neuron. We then took the mean absolute value of responses to that direction (relative to baseline), and plotted these values as a function of sensory condition for each cell (*Figure 1E*). Airflow responses were largest in two cell types targeting the ventral layers of the FB and the third compartment of the NO: P-F$_1$N$_3$ and P-F$_2$N$_3$. However, the cross-fly mean airflow response was larger than the mean stripe response in all cell types except P-F$_3$N$_{2v}$ and P-EN1. This difference was significant for half of the cell types surveyed: P-F$_1$N$_3$, P-F$_2$N$_3$, P-F$_3$N$_{2d}$, and P-EN2. The strength of sensory tuning tended to vary with raw response magnitude, with robustly responsive cell types also showing the strongest directional preferences (*Figure 1—figure supplement 2*). As such, many of the neurons we recorded were tuned for the direction of airflow (*Figure 1H*).

In contrast to this strong directional preference in the airflow condition, tuning strength in the visual condition was relatively weak across recorded cell types (*Figure 1—figure supplement 2*). However, visual tuning was not completely absent – for example, P-EN1 neurons displayed modest visual tuning, in agreement with previous results (*Turner-Evans et al., 2017*; *Green et al., 2017*). To ensure that our visual stimulus was sufficient to evoke responses in the CX, we recorded a small number of E-PG neurons, which are known to be tuned for both stripe and airflow direction (*Seelig and Jayaraman, 2015*; *Green et al., 2017*; *Okubo et al., 2020*). Recorded E-PGs showed directional tuning to the stripe (*Figure 1—figure supplement 3*), indicating that our stimulus can evoke tuned visual responses.

Adding odor to the airflow stream had only mild effects on CX columnar neuron responses. Odor slightly reduced the airflow responses of P-F$_1$N$_3$ and P-F$_2$N$_3$ (*Figure 1G*), although this difference was only significant for P-F$_1$N$_3$. Some single cells also showed increases or decreases in spiking activity in the odor condition (*Figure 1F*). While response magnitudes no doubt depend on stimulus intensity (airflow velocity, stripe contrast, and odor concentration), we know that our visual and airflow cues elicit orienting responses of approximately equal magnitude in a flight simulator (*Currier and Nagel, 2018*). Similarly, our odor cue produces robust orientation changes in walking flies (*Álvarez-Salvado et al., 2018*). Therefore, these responses reflect differential neural encoding of stimuli with similar behavioral relevance.

Broadly, our survey suggested that pairs of columnar cells innervating the same NO compartment share similar sensory responses. In particular, 'ventral P-FNs', which receive input in layer 3 of the NO and ventral FB layers, had large airflow responses, showed olfactory suppression, and preferred ipsilateral airflow (*Figure 1H*). 'Dorsal P-FNs', which receive input in layer 2 of the NO, and 'P-ENs', which receive input in layer 1 of the NO, both displayed modest sensory responses and ipsilateral

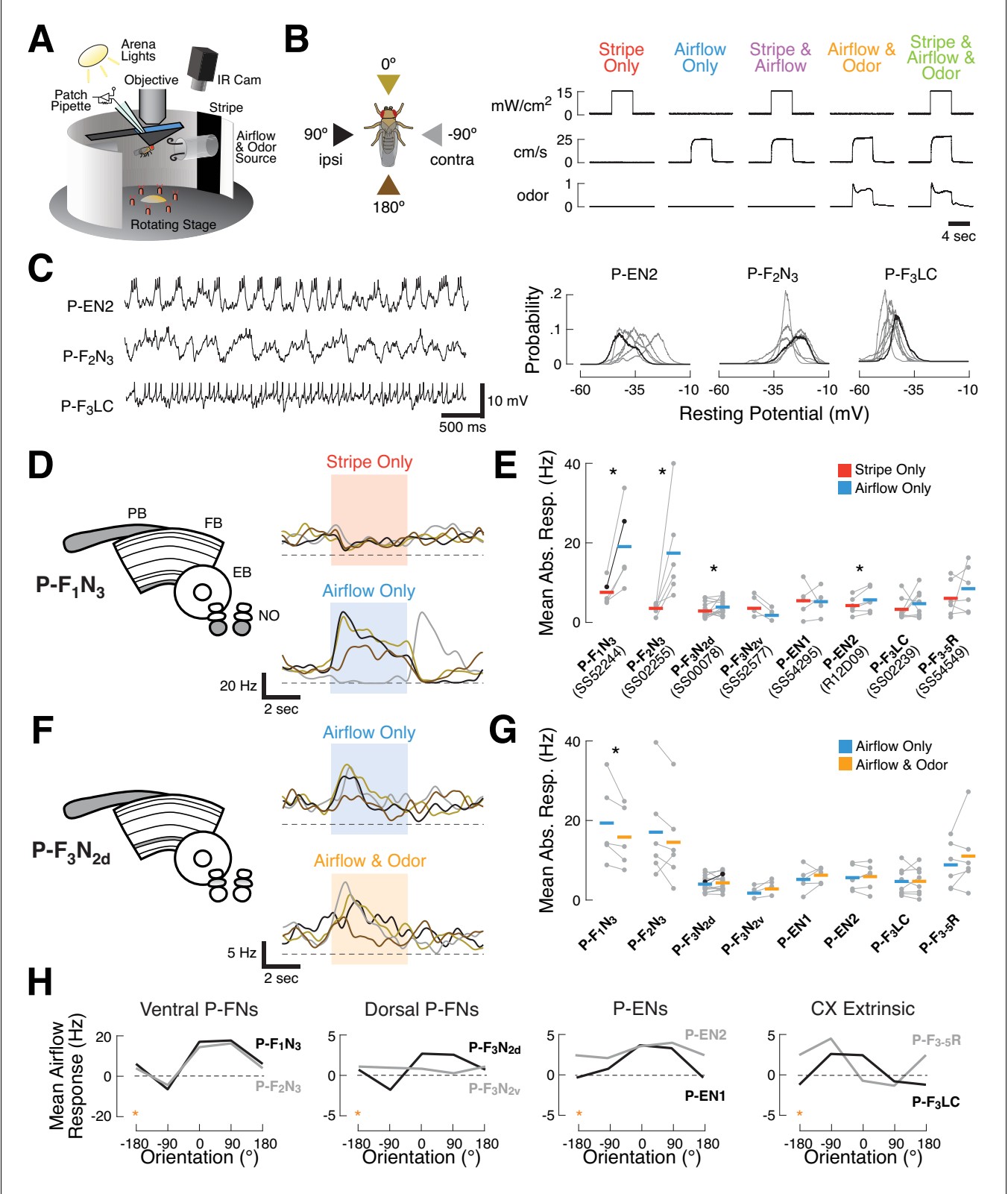

**Figure 1.** Sensory responses and preferred airflow direction vary across CX columnar cell types. (**A**) Experimental preparation. We targeted single neurons for patching using cell type-specific expression of GFP. Flies were placed in an arena equipped with rotatable stimulus delivery and live imaging of behavior. All data shown are from awake non-flying animals. (**B**) Stimulus details. Left: cue presentation directions. Front (0°, gold), rear (180°, brown), ipsilateral (90°, black), and contralateral (−90°, gray) to the recorded neuron. Right: stimulus validation. Each plot shows measurements from a

*Figure 1 continued on next page*

*Figure 1 continued*

photodiode (top), anemometer (middle), and photo-ionization detector (PID, bottom). PID units are arbitrary. The five stimulus combinations were: a high contrast stripe illuminated by 15 mW/cm2 ambient lighting (red), a 25 cm/s airflow stream (blue), stripe and airflow together (purple), airflow and 20% apple cider vinegar together (orange), and all three modalities simultaneously (green). Each trace is 12 s long. Simultaneous cues were presented from the same direction. (C) Rhythmic and tonic baseline activity in a subset of CX columnar neuron types. Left: raw membrane potential over time for three example neurons. P-EN2 and P-F$_2$N$_3$ show rhythmic activity at different frequencies, while P-F$_3$LC fires tonically at rest. Right: resting membrane potential probability distributions for each recorded neuron of the types shown (gray). Example neurons in black. Rhythmic neurons exhibit broad distributions, while tonic neurons show tight distributions. See also *Figure 1—figure supplement 1*. (D) Left: CX neuropils innervated by P-F$_1$N$_3$ neurons (gray). PB, protocerebral bridge; FB, fan-shaped body; EB, ellipsoid body; NO, noduli. Right: PSTHs for a single P-F$_1$N$_3$ neuron. Each trace represents the mean of four presentations of stripe alone (red, top) or airflow alone (blue, bottom) from one direction. Colors representing different directions as illustrated in (B). Colored boxes indicate the 4 s stimulus period. Dashed line indicates 0 Hz. (E) Responses to airflow (blue) versus stripe (red) for each neuron type. Gray dots indicate the mean spiking response of each cell (1 s stimulus minus 1 s baseline) to four trials from the direction producing the strongest response (see Materials and methods). Colored bars: mean across cells. The example P-F$_1$N$_3$ neuron from (D) is shown in black. Significant differences (p<0.05 by sign-rank test) between modalities are marked with an asterisk. For additional detail, see Fig. S2. (F) Left: CX neuropils innervated by P-F$_3$N$_{2d}$ neurons. Right: PSTHs for a single P-F$_3$N$_{2d}$ neuron. Each trace represents the mean of four presentations of airflow alone (blue, top) or airflow and odor together (orange, bottom) from one direction. Plot details as in (D). (G) Responses to odorized airflow (orange) versus airflow alone (blue) for all cell types recorded. Asterisk: odor significantly reduces the response of P-F$_1$N$_3$. Plot details as in (E). For additional detail, see *Figure 1—figure supplement 2*. (H) Mean airflow response across cells as a function of airflow direction for each cell type. Cell types are plotted in groups of two (gray, black) according to broad anatomical similarities. Note different vertical axis scales. Data at −180° is replotted from 180° for clarity (orange stars). For additional detail, see *Figure 1—figure supplement 2*.

The online version of this article includes the following figure supplement(s) for figure 1:

**Figure supplement 1.** Baseline activity characterization for recorded cell types.
**Figure supplement 2.** Summary of sensory responses across CX cell types.
**Figure supplement 3.** Tuned visual responses in E-PGs.

airflow tuning in most cases. In contrast, the 'CX extrinsic' columnar neurons we recorded, which target neuropils outside the CX, both preferred contralateral airflow. Thus, our survey suggests that sensory responses in CX neurons vary according to their input neuropils.

## Multi-sensory cues are summed in CX neurons, with some layer-specific integration variability

If CX compartments process unique combinations of sensory signals, we reasoned that neurons targeting different layers might also integrate multi-sensory signals in distinct ways. To understand the computational principles that govern cue integration in our recordings, we first compared multi-sensory responses to the sum of single modality responses (*Figure 2A*). Summation is a simple circuit principle that is naturally achieved by an upstream neuron or neurons passing multimodal cues to a single downstream cell. For each cell type, we found the mean response to the airflow-plus-stripe condition for each cue direction. We then plotted these multi-sensory responses against the sum of

**Table 1.** Intrinsic properties of surveyed neuron types.

| Cell type | Driver line | *N* | Resting potential (mV) | Input resistance (GΩ) | Osc. freq. (Hz) |
|---|---|---|---|---|---|
| P-F$_1$N$_3$ | SS52244 | 6 | −18.0 ± 1.0 | 6.21 ± 0.52 | 4 |
| P-F$_2$N$_3$ | SS02255 | 6 | −22.0 ± 1.3 | 4.86 ± 0.50 | 2 |
| P-F$_3$N$_{2d}$ | SS00078 | 14 | −30.9 ± 0.9 | 2.58 ± 0.24 | - |
| P-F$_3$N$_{2v}$ | SS52577 | 4 | −31.8 ± 3.5 | 6.00 ± 0.50 | - |
| P-EN1 | SS54295 | 4 | −32.9 ± 2.1 | 2.52 ± 0.29 | 3 |
| P-EN2 | R12D09 | 6 | −30.1 ± 1.9 | 3.30 ± 0.29 | 3 |
| P-F$_3$LC | SS02239 | 8 | −39.7 ± 0.8 | 2.75 ± 1.11 | - |
| P-F$_{3-5}$R | SS54549 | 6 | −26.3 ± 1.3 | 7.89 ± 0.51 | - |
| E-PG | SS00090 | 4 | −29.4 ± 1.5 | 2.30 ± 0.35 | - |

Resting potential, input resistance, and characteristic oscillatory frequency are shown for each recorded cell type. Values represent the cross-fly mean +/- SEM. See also *Figure 1—figure supplement 1*.

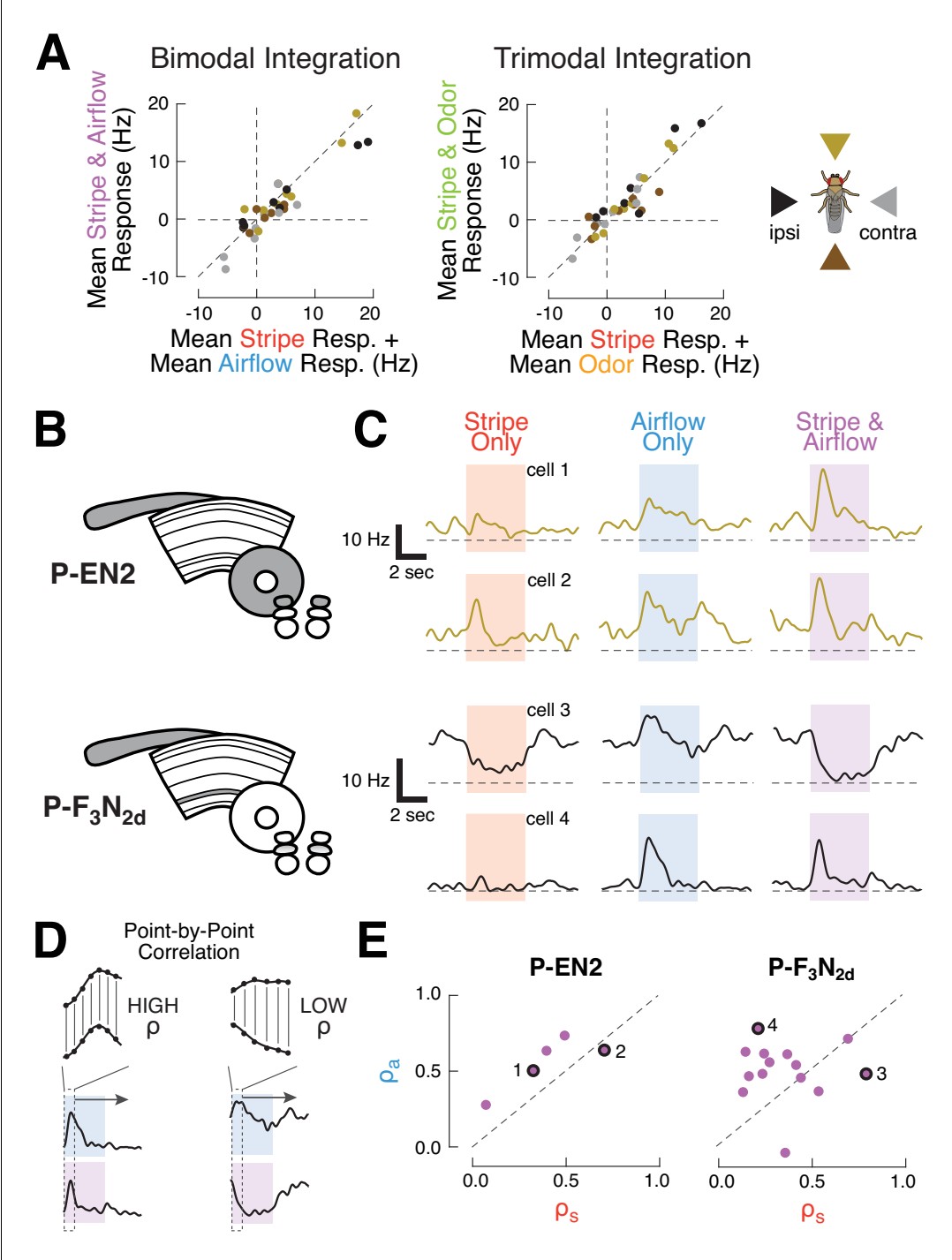

**Figure 2.** CX columnar neurons sum inputs from different modalities on average, but show diverse integration strategies at the level of single cells. (**A**) Summation of multimodal cues. Left: mean spiking response to stripe and airflow together versus sum of mean stripe alone and airflow alone responses. Each point represents the response of one cell type to cues from one direction. Right: mean spiking response to stripe and airflow and odor versus sum of mean stripe alone response and mean airflow and odor response. Colors indicate cue direction (far right). Data falling along the diagonals indicate perfectly weighted summation. (**B**) CX neuropils innervated by example cell types P-EN2 and P-F$_3$N$_{2d}$. (**C**) PSTHs of two neurons from each cell type. Curves represent mean firing rate across four trials of each stimulus from a single direction. Colored boxes indicate the four second stimulus period. Dashed lines indicate 0 Hz. Top: example P-EN2 neurons responding to frontal cues. In both cases, the multi-sensory response (purple) is a weighted sum of the single modality responses (red, blue). Bottom: P-F$_3$N$_{2d}$ neurons responding to ipsilateral cues. In one cell (cell 3) the stripe response dominates the multi-sensory response, while in the other (cell 4) the airflow response dominates. (**D**) Correlation method for computing response similarity. We computed a point-by-point correlation between the mean baseline-subtracted firing rate timecourses of multi-sensory (airflow

*Figure 2 continued on next page*

*Figure 2 continued*

and stripe together) responses and responses to a single modality (airflow alone, or stripe alone), across all stimulus directions. Similar traces result in high correlation coefficient (ρ). (E) Correlation coefficients (calculated as in D) of the multimodal response (stripe and airflow together) to each single modality response (airflow alone, $\rho_a$, or stripe alone, $\rho_s$). Data along the diagonal indicates that the multi-sensory response is equally similar to the stripe alone and airflow alone responses, a hallmark of summation. Data off diagonal indicates that one modality dominates the multi-sensory response. The four example cells from (C) are labeled with numbers and black rings. P-EN2 neurons consistently sum stripe and airflow responses (top), while P-F$_3$N$_{2d}$ neurons integrate with greater diversity (bottom).

The online version of this article includes the following figure supplement(s) for figure 2:

**Figure supplement 1.** Characterization of multi-sensory integration for recorded cell types.

the mean airflow response and the mean stripe response from the same directions (*Figure 2A*, left). When we plotted these measures for each cell type, we found that most points fell on or near the diagonal, indicating that multi-sensory responses are, on average, approximately equal to the sum of single modality responses. This trend of near-perfect summation was also true when all three modalities were presented simultaneously (*Figure 2A*, right). These results suggest that poly-modal integration generally proceeds via a summation principle, at least for the stimuli presented here, which were always presented from the same direction.

Because of the diverse sensory responses observed in our survey, we wondered whether integration principles may also differ across individual members of a single cell type. To answer this question, we evaluated stripe and airflow integration for single cells (*Figure 2B–E*). Broadly, we found that integration diversity was small for some cell types, but large for others. P-EN2 neurons, for example, showed remarkably consistent summation. Single neuron spiking responses to multi-sensory cues strongly resembled the responses to single modality cues across all P-EN2s (*Figure 2C*, top). To evaluate summation in a scale-free manner, we found the correlation coefficient between the mean response timecourse to the multi-sensory condition (airflow and stripe together) and each of the single modality conditions (airflow alone or stripe alone, *Figure 2D*). To do this, we concatenated each cell's mean spiking responses to stimuli presented from different directions (−90°, 0°, 90°, 180°), with the baseline period removed. We then took the point-by-point correlation between the concatenated multi-sensory response and the single modality responses, which yielded a pair of correlation coefficients ($\rho_a$ for airflow and $\rho_s$ for stripe). A coefficient of 1 indicates that the multi-sensory and single modality data varied over time in perfect synchrony. Conversely, a coefficient closer to 0 indicates that these signals did not vary together. When we plotted these coefficients against one another (*Figure 2E*, but also see *Figure 2—figure supplement 1*), we found that the P-EN2 data lie along the diagonal, indicating that the stripe and airflow responses equally resemble the multi-sensory response for each neuron.

In contrast, P-F$_3$N$_{2d}$ neurons show much greater integration diversity. While some cells displayed multi-sensory activity that was dominated by the stripe response, others were dominated by the airflow response (*Figure 2C*, bottom). Indeed, the correlation coefficients for P-F$_3$N$_{2d}$ reveal a full range of modality preferences (*Figure 2E*, right). We did not observe any obvious relationship between single neuron anatomy and the method of integration used by that cell, although this result might reflect a limitation of our stimuli – for example, if these neurons were preferentially tuned to stimuli at a particular phase offset. Thus, while summation appears to govern P-F$_3$N$_{2d}$ integration on average, individual neurons show an array of sensory integration strategies. This trend of summation on average, but diversity at the single cell level, was found for many of the surveyed cell types (*Figure 2—figure supplement 1*). These results suggest that CX neurons integrate multi-modal sensory cues with compartment-specific variability.

## Ventral P-FN airflow responses are organized as orthogonal basis vectors, rather than as a map or compass

At the conclusion of our survey, ventral P-FNs (P-F$_1$N$_3$ and P-F$_2$N$_3$) stood out as possessing the most robust sensory responses, prompting us to examine the activity of P-F$_2$N$_3$ in greater detail (*Figure 3A–G*). These cells had resting membrane potentials between −30 and −25 mV, and input resistances around 3 GOhm (Fig. S1). P-F$_2$N$_3$s showed strong spiking responses to single presentations of ipsilateral airflow, and active inhibition followed by offset spiking during single presentations

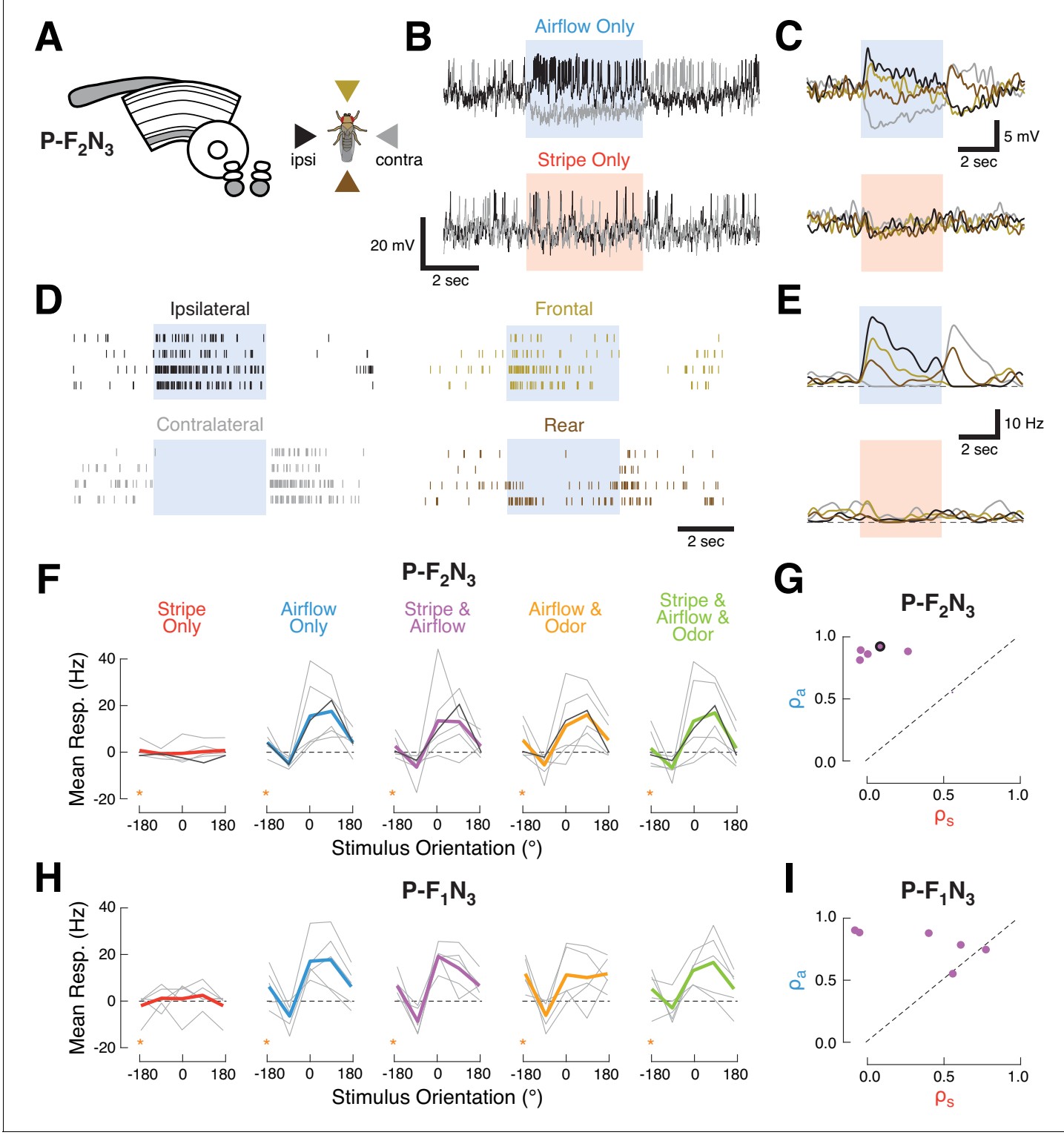

**Figure 3.** Ventral P-FNs selectively respond to directional airflow. (**A**) Left: CX neuropils innervated by P-F₂N₃. Right: color key for directional stimuli. (**B**) Example trials from a single P-F₂N₃ neuron. Raw membrane potential for single presentations of airflow alone (blue, top) or stripe alone (red, bottom) for ipsilateral (black) and contralateral (gray) directions. Colored box indicates 4 s stimulus period. Baseline Vₘ = −28 mV. (**C**) Average Vₘ (over four trials) for the example neuron shown in (**B**). Colors represent directions as shown in (**A**). Stimulus period represented as in (**B**). (**D**) Spike response rasters for the example neuron shown in (**B**). Colors and stimulus period as in (**C**). (**E**) PSTHs for the example neuron in (**B**). Colors and stimulus period as in (**C**). (**F**) P-F₂N₃ direction tuning for each cue set showing that responses to airflow are not modulated by other modalities. Mean spiking response minus

*Figure 3 continued on next page*

*Figure 3 continued*

baseline for each recorded cell as a function of stimulus direction (gray lines). The example neuron in (**B–E**) is shown in black. Mean tuning across cells shown in thick colored lines. Data at −180° is replotted from 180° for clarity (orange stars). (**G**) Similarity (as in *Figure 2C*) of P-F$_2$N$_3$ multi-sensory (stripe + airflow) responses to airflow alone and stripe alone. Response to stripe + airflow is highly similar to airflow alone. Example neuron marked in black. (**H**) Direction tuning for the second type of ventral P-FN, P-F$_1$N$_3$. Note that odor subtly inhibits airflow-evoked responses (as shown in *Figure 1G*). Data at −180° is replotted from 180° for clarity (orange stars). (**I**) Same as (**G**), but for P-F$_1$N$_3$.

of contralateral airflow (*Figure 3B&D*). Spiking activity in response to ipsilateral and contralateral airflow was relatively consistent from trial to trial, while frontal and rear airflow elicited more diverse responses relative to baseline on each trial (*Figure 3D*). On average, P-F$_2$N$_3$ neurons showed graded membrane potential and spiking responses to airflow, but not to the stripe (*Figure 3C&E*).

To assess how additional sensory modalities modulate this directional airflow tuning, we plotted the spiking response as a function of stimulus orientation for each combination of cues (*Figure 3F*). We found that mean tuning across the population did not change when the stripe, odor, or both, were added to airflow. Each P-F$_2$N$_3$ neuron showed a large airflow correlation coefficient and a small visual coefficient (*Figure 3G*), indicative of multi-sensory responses that strongly resemble the airflow-only response. P-F$_1$N$_3$ sensory activity was similar, except for the olfactory suppression noted above (*Figure 3H&I*).

Like many other columnar cell types, individual ventral P-FNs target one CX column, with cell bodies in each hemisphere collectively innervating the eight outer columns of the ipsilateral PB and all eight columns of the FB (*Wolff et al., 2015*; *Hulse et al., 2020*). We next asked whether neurons innervating different columns show distinct directional tuning, as has been previously observed for polarized light cues (*Heinze and Homberg, 2007*) and for visual landmarks and airflow in E-PG compass neurons (*Green et al., 2017*; *Fisher et al., 2019*; *Okubo et al., 2020*). To address this question, we recorded from a larger set of P-F$_2$N$_3$ neurons while explicitly attempting to sample from a range of CX columns (*Figure 4A*). For this experiment, we presented airflow from eight directions and omitted other sensory modalities. To identify the FB columns targeted by our recorded neurons, we filled each cell with biocytin and visualized its anatomy after the recording session (*Figure 4B&C*).

Surprisingly, we found that all left-hemisphere P-F$_2$N$_3$ neurons responded strongly to airflow presented from the front-left (between 0° and 90°, ipsilateral) and were inhibited by airflow from the rear-right (between −90° and −180, contralateral), regardless of the column they innervated. One right-hemisphere neuron was excited by airflow from the front-right and inhibited by airflow from the rear-left. Thus, all P-F$_2$N$_3$ neurons show a preference for airflow presented ipsi-frontal relative to the hemisphere of their cell bodies (*Figure 4D&E*), with peak tuning around approximately 45° ipsilateral.

We did notice some subtle differences in tuning that varied with column. The spiking response to frontal airflow (0°) was strongest in the ipsilateral-most column 1, and was weaker in more contralateral columns (*Figure 4F*). Membrane potential responses to contralateral also airflow varied by column, with contralateral columns exhibiting the greatest inhibition relative to baseline (*Figure 4G, H*). P-F$_2$N$_3$ neurons also showed temporally diverse airflow responses (*Figure 4I*). Some neurons showed sustained activity during the stimulus period, while others displayed only transient responses to airflow presentation. Temporal responses were not reliably organized by column (*Figure 4J,K*) but might instead reflect different behavioral states of the animal.

These data support two conclusions. First, ventral P-FNs respond primarily to airflow, and not to the other stimuli presented in our cue set. Second, airflow tuning across P-F$_2$N$_3$ neurons is not organized as a map of airflow direction, but is instead clustered around two directions approximately 45° to the left and right of the fly midline. This is reminiscent of the organization of optic flow responses in TN neurons of the bee (*Stone et al., 2017*), which have been proposed to act as basis vectors for computing movement of the animal through space. Since each FB column is innervated by ventral P-FNs with cell bodies in both hemispheres, each column receives two orthogonal airflow signals that could be used to construct tuning to a variety of airflow directions in downstream neurons.

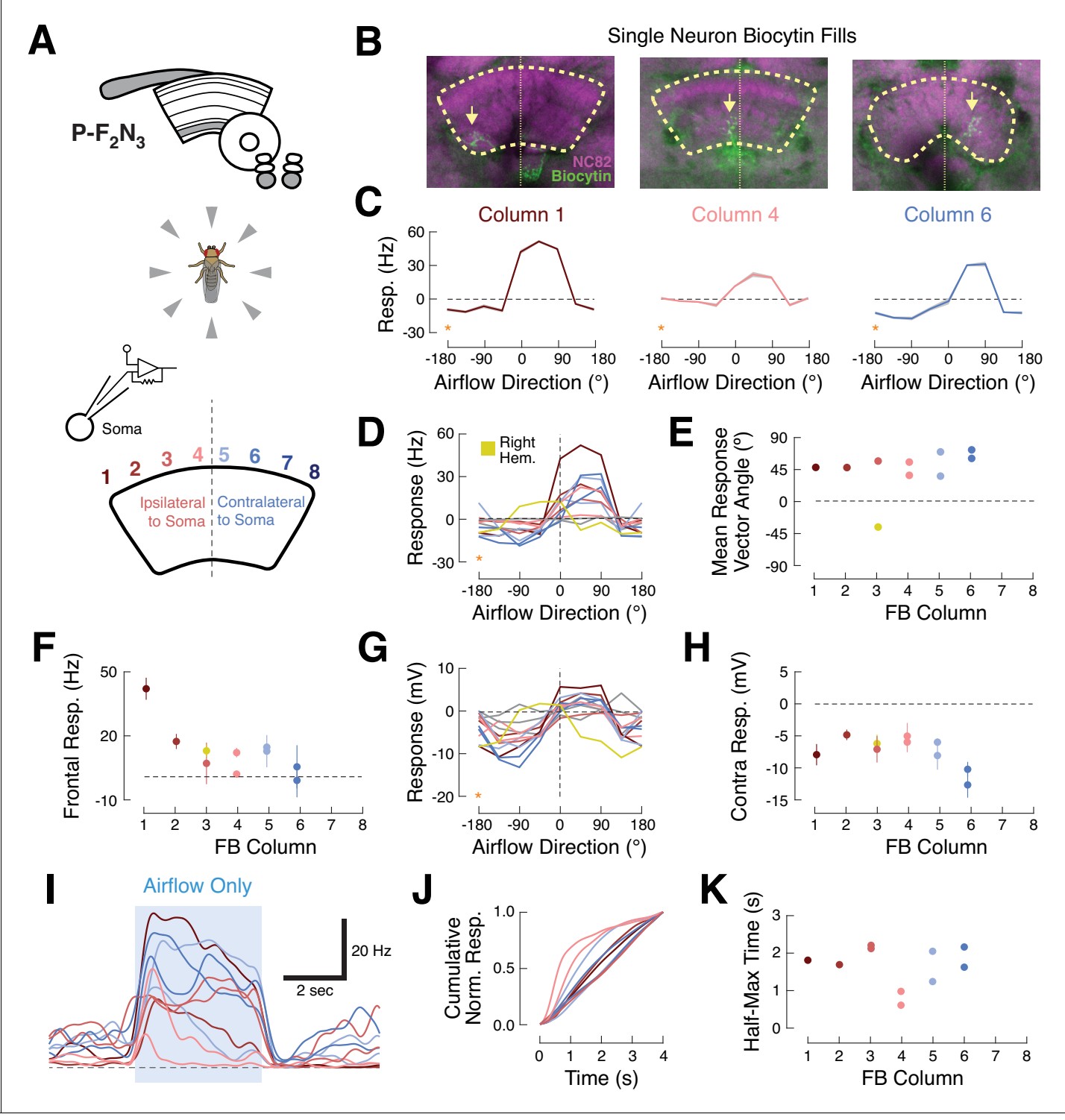

**Figure 4.** Ventral P-FNs exhibit similar ipsilateral airflow tuning across CX columns. (**A**) Top: CX neuropils innervated by P-F$_2$N$_3$. Bottom: experimental setup. We presented airflow from eight directions and identified the column innervated by each patched neuron by filling the cell with biocytin. (**B**) Biocytin fills (green) for three example cells innervating columns 1 (left), 4 (middle), and 6 (right). Yellow arrows indicate FB portions of fills. Neuropil in magenta. Thick dashed line indicates the borders of the FB and thin dotted line shows the midline. (**C**) P-F$_2$N$_3$ airflow tuning is similar across FB columns. Mean +/- SEM spiking response as a function of airflow direction for the three example cells shown in (**B**). Data at −180° is replotted from 180° for clarity (orange stars). Colors reflect innervated column, as in (**A**). (**D**) Mean spiking response as a function of airflow direction for all recorded P-F$_2$N$_3$ neurons. Colors as in (**A**). A single right-hemisphere neuron is shown in yellow. Gray curves indicate cells for which no anatomy data could be recovered. Data at −180° is replotted from 180° for clarity (orange star). (**E**) Mean response vector angle as a function of column for each recorded

*Figure 4 continued on next page*

*Figure 4 continued*

neuron. Colors as in (D). (F) Mean +/- SEM spiking response to frontal airflow as a function of column. Colors as in (A). (G) Mean membrane potential response as a function of airflow direction for each neuron. Data at −180° is replotted from 180° for clarity (orange star). Colors as in (D). (H) Mean +/- SEM membrane potential response to contralateral airflow as a function of column. Colors as in (D). (I) Timecourse (PSTH) of airflow responses to ipsilateral airflow for the same P-F$_2$N$_3$ neurons (average of 4 trials). Blue box indicates 4 s stimulus. Colors as in (A). (J) Cumulative normalized response for each neuron during the 4 s stimulus, normalized to its mean integrated response. Transient responses show fast rise times and tonic responses show slower rise times. Colors as in (A). (K) Time to half-max (a measure of response transience) as a function of column. Colors as in (A).

## Ventral P-FNs likely inherit their airflow tuning from the Lateral accessory lobe (LAL)

What is the source of the airflow signals in ventral P-FNs? Neurons sensitive to airflow direction have recently been identified in both the antler (ATL, *Suver et al., 2019*), and the lateral accessory lobe (LAL, *Okubo et al., 2020*). We identified two candidate populations that might carry airflow signals to ventral P-FNs. Using trans-tango experiments, we found that a group of ventral FB neurons (vFBNs) receive input in the antler and appear to be presynaptic to ventral PFNs (*Figure 5—figure supplement 1*). The *Drosophila* hemibrain connectome (*Scheffer et al., 2020*) indicates that P-F$_2$N$_3$ neurons (PFNa in the hemibrain) receive prominent input from LNa neurons (LAL-NO(a) neurons, *Wolff and Rubin, 2018*) that receive input in the LAL and project to the third compartment of the NO. To assess whether either of these groups of neurons carry tuned airflow signals, we recorded from vFBNs and performed 2-photon calcium imaging using GCaMP6f from LNa neurons (*Figure 5B*). LNa somata were not accessible for electrophysiology in our preparation.

We found that LNa neurons, but not vFBNs, possessed directionally tuned airflow responses (*Figure 5C*). Like ventral P-FNs, LNa neuron activity was strongly modulated by wind direction but not by the presence of odor (*Figure 5C*). Since all of the strongly airflow-tuned ventral P-FNs receive input in the third compartment of the NO in one hemisphere, this finding could explain why all ventral P-FNs in one hemisphere share similar sensory tuning. Right-hemisphere LNas are connected to left-hemisphere ventral P-FNs, and vice-versa (*Figure 5A*). When we specifically compared the tuning of LNa neurons to synaptically connected ventral P-FNs, we found that LNa tuning was inverted (*Figure 5C*), suggesting that LNa neurons are inhibitory. Together, these results suggest that tuned airflow responses in ventral P-FNs are likely inherited from airflow-sensitive populations in the LAL (WL-L neurons, *Okubo et al., 2020*), although silencing experiments will be required to directly test the contribution of LNa and WL-L neurons to ventral P-FN sensory tuning.

## Ventral P-FNs are required to orient to airflow in tethered flight

Finally, we wondered whether ventral P-FNs play a role in orientation to airflow. We addressed this question with a previously designed closed-loop flight simulator that uses an infrared camera to monitor the fictive turning of a tethered animal flying in the dark (*Figure 6A*). This turn signal drives rotations of an airflow tube, allowing flies to control their orientation with respect to that flow. In previous experiments, we observed that flies prefer to orient away from the source of flow (*Currier and Nagel, 2018*).

We first asked whether silencing ventral P-FNs (P-F$_1$N$_3$ and P-F$_2$N$_3$) impairs normal airflow-based orienting. We compared behavior in flies where these neurons were silenced with Kir2.1 to control flies where Kir2.1 was driven by an *empty-GAL4* cassette. Consistent with our previous results, control flies adopted stable orientations away from the airflow source (*Figure 6C–F*). When we calculated the mean orientation vector for each control fly, we found that they all preferred orientations roughly opposite the flow source, near 180° (*Figure 6D*). When either type of ventral P-FN was silenced with Kir2.1, flies displayed partially impaired orientation ability (*Figure 6D*). These groups showed increased orienting toward the flow (*Figure 6E*) and reduced orientation stability (*Figure 6F*) compared to controls, although both effects were moderate.

Given the similar sensory responses of P-F$_1$N$_3$ and P-F$_2$N$_3$, we reasoned that they might serve overlapping roles. We therefore sought a driver line that labeled both classes of ventral P-FN, and identified *R44B10-GAL4* as one such line with minimal off-target expression (*Figure 6B* and *Figure 6—figure supplement 1*). Silencing *R44B10-GAL4* neurons with Kir2.1 produced a more severe phenotype (*Figure 6C*). Compared to controls, these flies showed less stable orientation to airflow (*Figure 6D*), spent significantly more time oriented toward the airflow source (*Figure 6E*), and had

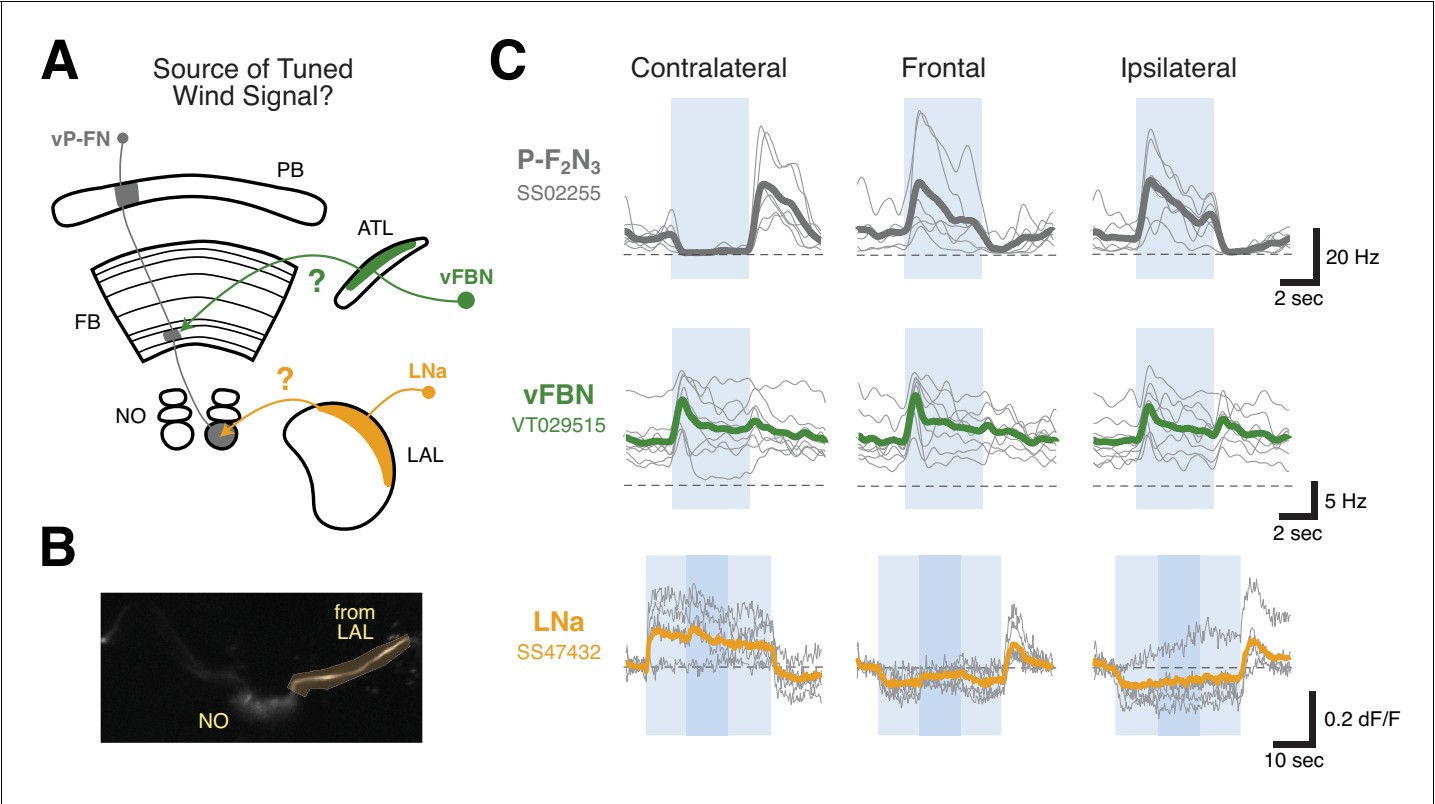

**Figure 5.** LNa neurons are a likely source of airflow signals in ventral P-FNs. (**A**) Experimental framework. Neurons with tuned airflow responses have recently been identified in the Antler (ATL) and Lateral Accessory Lobe (LAL). We recorded from vFBN (green) and LNa (orange) neurons to assess which input pathway might carry tuned airflow signals. (**B**) Max projection of the NO region of the *SS47432 > UAS-GCaMP6f* line used to record LNa calcium activity using tdTomato signal. Imaging ROI, highlighted in orange, is the neurite of one LNa neuron in one hemisphere that connects the LAL and NO. (**C**) Ventral P-FN airflow tuning is likely inherited from the LAL. Mean firing rate (top two rows) or dF/F (bottom row) as a function of time is plotted for each fly (thin gray lines). Cross-fly mean activity is plotted as thick colored lines. Responses to airflow presented contralaterally (−90°, left column), frontally (0°, middle column), and ipsilaterally (90°, right column) are shown. Directions (ipsi, contra) are relative to the hemisphere of connected ventral P-FN cell bodies. vFBNs responded to airflow, but were not sensitive to airflow direction. LNa neurons showed strong directional tuning for airflow that is sign-inverted with respect to ventral P-FN activity. Blue boxes represent stimulus period (4 s for top two rows, 30 s for bottom row), while dashed lines indicate 0 Hz or dF/F. Darker blue region in the bottom row represents a 10 s period when 10% apple cider vinegar was injected into the airstream (while maintaining constant airflow velocity). Odor did not have a statistically significant impact on LNa activity. Colors as in (**A**).

The online version of this article includes the following figure supplement(s) for figure 5:

**Figure supplement 1.** Trans-tango of VT029515 (vFBN).

reduced fixation strength (*Figure 6F*). Collectively, these results suggest that ventral P-FNs are required for normal orientation to airflow. We attempted to broadly silence ventral P-FNs using several other genotypes (*15E12-GAL4*, *67B06-GAL4*, and *20C08-GAL4*), however none of these flies were viable when crossed to *UAS-Kir2.1*.

E-PG neurons also respond robustly to directional airflow cues (*Okubo et al., 2020*). Thus, we wondered whether silencing these cells would also impair orientation to airflow. In contrast to our experiments with ventral P-FNs, we found that silencing E-PGs did not disrupt orientation to airflow (*Figure 6C–F*). In flies with silenced E-PGs, both the time spent orienting toward the airflow, and orientation stability, were indistinguishable from controls (*Figure 6E–F*). The split-GAL4 line used to silence E-PGs was similarly sparse to the lines used to silence P-F$_1$N$_3$ and P-F$_2$N$_3$ alone, strengthening the conclusion that ventral P-FNs specifically play a functional role in orientation to airflow.

To obtain more insight into the mechanism by which ventral P-FNs control orientation, we pseudo-randomly introduced six stimulus perturbations throughout each 20 min closed-loop testing session (*Figure 7A*). These were long (2 s) or short (150 ms) pauses in airflow ('airflow off'), and long

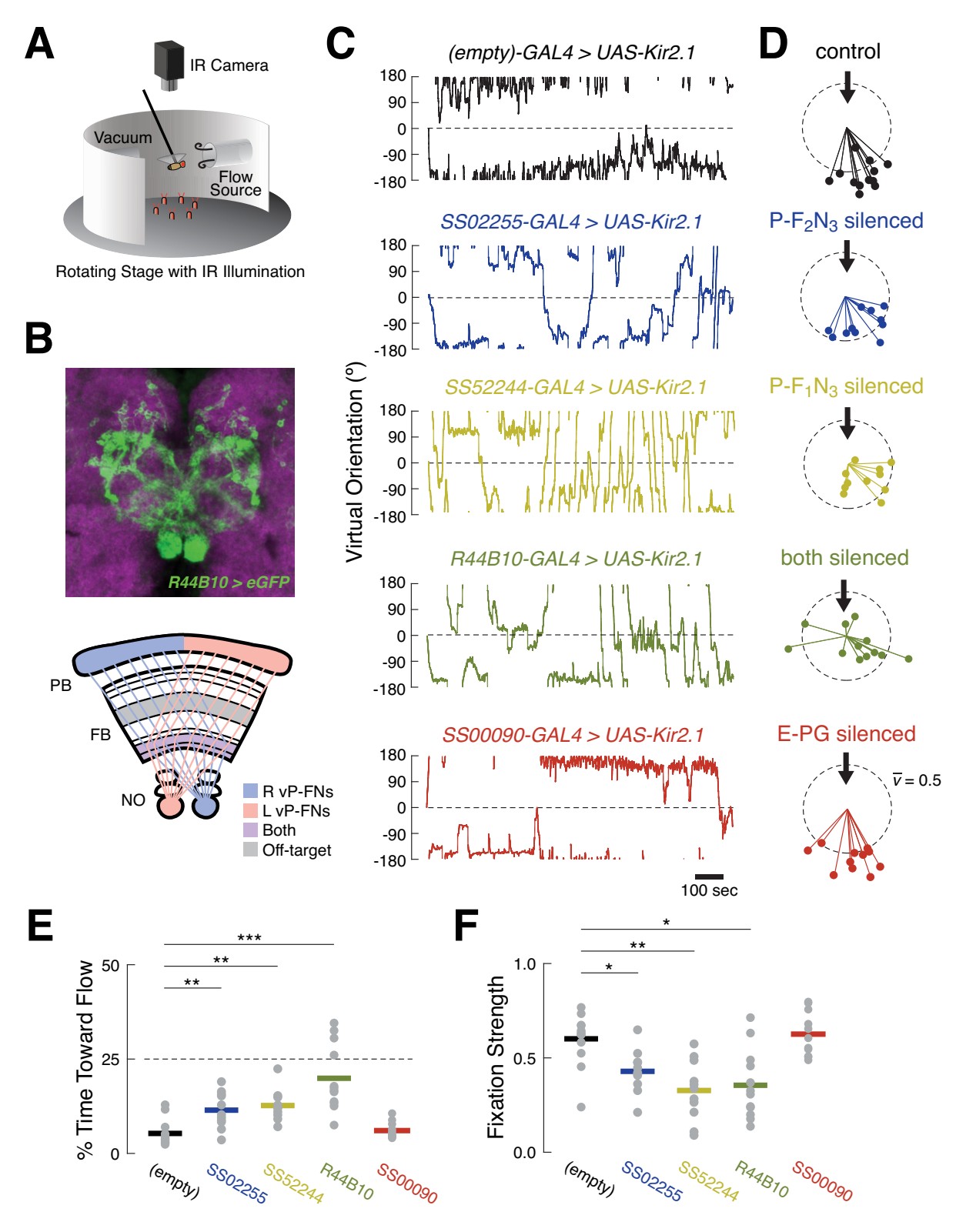

**Figure 6.** Silencing ventral P-FNs disrupts orientation to airflow. (A) Schematic of flight simulator arena. Rigidly tethered flying flies orient in closed-loop with an airflow stream. Infrared illumination is used to track wingbeat angles which drive airflow rotation. The arena was otherwise in darkness. Modified from *Currier and Nagel, 2018*. (B) Anatomy of *R44B10-GAL4*, a driver line that targets both P-F$_1$N$_3$ and P-F$_2$N$_3$. Top: maximum z-projection of R44B10 driving *10XUAS-Kir.21-eGFP* (green). Neuropil in magenta. Bottom: schematic of CX neuropils labeled by *R44B10-GAL4*. Right-hemisphere ventral

*Figure 6 continued on next page*

*Figure 6 continued*

P-FNs (blue) target the right half of the PB, the entire FB, and the left NO. Left-hemisphere ventral P-FNs (red) target the left half of the PB, the entire FB, and the right NO. Neurons from both hemispheres target all columns of FB layers 1 and 2 (purple). *R44B10-GAL4* also labels non-P-FN neurons in a FB layer ('off target,' gray). Whole-brain expression in Fig. S5. (C) Orientation over time for example flies of each genotype. Control flies (*empty-GAL4 > UAS-Kir2.1*, black) fixate orientations away from the airflow sources. This fixation is reduced in flies with P-F$_2$N$_3$ (blue), P-F$_1$N$_3$ (yellow), or both (dark green) silenced. Flies with E-PGs silenced (red) show control-like orientation behavior. The entire 20 min testing period is shown for each fly. Airflow emanated from 0° (dashed line). (D) Stick-and-ball plots of mean orientation (ball angle) and fixation strength (stick length) for each fly tested in the airflow orienting paradigm. Fixation strength is the length of the mean orientation vector, which is inversely proportional to circular variance (see Materials and methods). Dashed circle corresponds to fixation strength of 0.5. All but one control fly, and all E-PG-silenced flies, showed fixation strengths near or above this value, while ventral P-FN-silenced flies displayed smaller fixation strengths. Thick arrow signifies the position and direction of the airflow stimulus (0°). Colors as in (C). (E) Percentage of time each fly (gray dots) oriented toward the flow source (between +45° and −45°), as a function of genotype. Horizontal bars indicate cross-fly means, with colors as in (C). Dashed line indicates the expected value for random orienting (chance). **p<0.01; ***p<0.001 (rank-sum test). (F) Fixation strength (as illustrated in (D)) for each fly as a function of genotype. A value of 1 indicates perfect fixation. Plot details as in (E). *p<0.05; **p<0.01 (rank-sum test).

The online version of this article includes the following figure supplement(s) for figure 6:

**Figure supplement 1.** Full central brain anatomy of *R44B10-GAL4.*

(63.36°) or short (14.44°) angular displacements of the airflow source to the left or right ('airflow slip'). In response to airflow pauses, control flies briefly turned toward the airflow source (*Figure 7B*). When the airflow resumed, flies once again turned away (see also *Currier and Nagel, 2018*). Overall turning in response to airflow pauses remained unchanged when both types of ventral P-FN were silenced (*Figure 7C*). These results suggest that, despite the abnormal orienting behavior shown by ventral P-FN-silenced flies, they are still able to detect airflow and determine its direction.

We next wondered if the poor orienting ability of ventral P-FN-silenced flies arises from motor deficits. However, we found no differences between the distributions of wingbeat angle differences for control and silenced flies (*Figure 7D*). Similarly, silencing ventral P-FNs did not change the distribution of integrated turn angles following airflow direction slips (*Figure 7E*). Together with our airflow pause data, these results suggest that silencing ventral P-FNs leaves both sensory and motor function intact.

In contrast, silencing ventral P-FNs dramatically impaired the selection of turns following airflow direction slips. In response to slips, control flies generally made corrective turns in the direction opposite the slip (*Figure 7F*, top). For rightward slips, control flies made turns to left, and for leftward slips, turns to the right. The duration and magnitude of the slip response varied with slip duration in control flies, such that turns following long slips (*Figure 7F*, left) were larger than turns following short slips (*Figure 7F*, right). On average, control flies' reactive turns corrected for 45% of the orientation change induced by long slips, and 90% of short slips (*Figure 7G*). Conversely, flies with both types of ventral P-FNs silenced (*R44B10-Gal4>UAS-Kir2.1*) showed no turning response, on average, to slips of any direction or duration (*Figure 7F&G*). When only one type of ventral P-FN was silenced, we observed a smaller reduction in mean slip responses. Silencing E-PGs did not disrupt slip correction (*Figure 7F&G*), as expected based on the orientation behavior. Together, these results suggest that ventral P-FNs may be specifically involved in generating an appropriate turning response following a change in the direction of airflow.

## Discussion

### Distinct sensory representations in different CX compartments

All animals make use of many different sensory cues to navigate through their environments. Insects and rodents use visual landmarks to return to remembered locations (*Ofstad et al., 2011*; *Collett et al., 2001*; *Etienne et al., 1990*). Odor cues are widely used to navigate toward sources of food or mates (*Baker et al., 2018*). Movements of the air or water are prominent cues for orientation and navigation across species (*Chapman et al., 2011*; *Alerstam et al., 2011*). How neural circuits are organized to process and combine these diverse cues is a fundamental question in neuroscience and evolution.

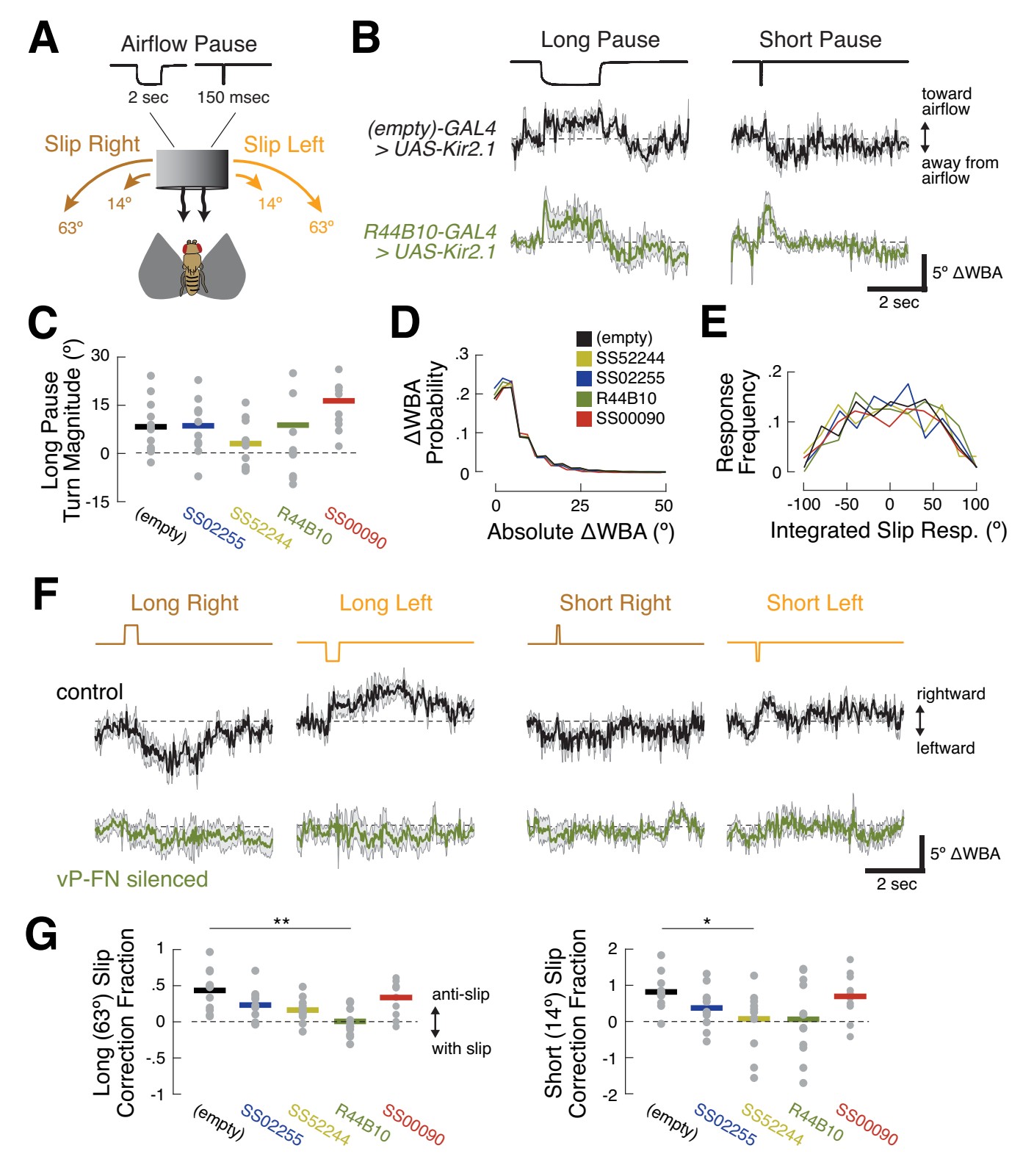

**Figure 7.** R44B10 neurons are required to convert airflow orientation changes into heading-appropriate turns. (**A**) Stimulus manipulations. Six manipulations were presented pseudo-randomly every 20 s during closed-loop flight: long wind pause (2 s); short wind pause (150 msec); short (14.44°) and long (63.36°) rightward slip of virtual orientation; and short and long leftward slip of virtual orientation. Slip velocity was 144 °/sec. (**B**) Responses to long and short airflow pauses in control flies (*empty-GAL4>UAS-Kir2.1*, black) and ventral P-FN-silenced flies (*R44B10-GAL4>UAS-Kir2.1*, dark green).

*Figure 7 continued on next page*

*Figure 7 continued*

Traces show mean +/- SEM difference in wingbeat angles (ΔWBA), a proxy for intended turning, for 120 trials across 12 files (10 repetitions per fly). In this plot, positive ΔWBA values indicate turns toward the airflow source and negative ΔWBA values indicate turns away from the airflow source. Dashed line represents no turning. (C) Mean ΔWBA (integrated over 2 s) in response to a long airflow pause for each fly (gray dots) of each genotype. Horizontal bars indicate cross-fly means. Positive ΔWBA values represent turns toward the airflow source. All groups are statistically indistinguishable by rank-sum test. (D) Probability distributions of ΔWBA values for control (black), P-F$_1$N$_3$-silenced (gold), P-F$_2$N$_3$-silenced (dark blue), all ventral P-FN-silenced (dark green), and E-PG silenced (red) flies. The distributions are statistically indistinguishable by KS-test. (E) Probability distributions of integrated slip responses for each genotype (colors as in (D)). Slip responses are integrated over 5 s of slip stimulus, with both leftward (negative) and rightward (positive) slips included. The distributions are statistically indistinguishable by KS-test. (F) Responses to orientation slips in ventral P-FN-silenced flies (*R44B10-GAL4>UAS-Kir2.1*, dark green) and control flies (*empty-GAL4>UAS-Kir2.1*, black). Each trace represents the mean +/- SEM of 120 trials across 12 files (10 repetitions per fly). In this plot, positive ΔWBA values indicate rightward turns and negative ΔWBA values indicate leftward turns. Traces show cross-fly mean +/- SEM with colors as in (B). (G) Fraction of slip displacement corrected by each fly (gray dots) of each genotype in response to long (left) and short (right) slips. Values represent mean integrated slip response divided by negative slip magnitude. Positive values indicate 'corrective' turns in the opposite direction of the slip. A correction fraction of 1 indicates that a fly steered the airflow direction to be identical before and after a slip trial. Note differing Y-axis scales for each plot. *p<0.05; **p<0.01 (rank-sum test).

Although previous studies have examined CX responses to a wide variety of sensory cues (*Heinze and Homberg, 2007*; *Weir and Dickinson, 2015*; *Ritzmann et al., 2008*; *Phillips-Portillo, 2012*; *Kathman and Fox, 2019*), it has remained unclear to what extent these responses are organized or specialized across CX compartments. By targeting specific cell populations using genetic driver lines, the present study supports the hypothesis that distinct cell types within the CX represent specific sensory cues. Ventral P-FNs exhibited directional responses to airflow that were much stronger than those of the other cell types in our survey. In our study, only E-PGs showed strong directional tuning to the visual landmark, although it is possible that our survey, which only examined responses to four cardinal directions, may have missed responses primarily tuned for directions 45° off from these. Finally, ventral P-FNs might show stronger responses to other sensory cues during closed-loop behavior, when heading signals from the E-PG compass system are engaged.

A striking finding of our study is that the same sensory cue may be represented in different formats in different CX compartments (*Figure 8*). For example, a recent study found a map-like representation of wind direction in E-PGs, with different directions systematically represented across columns (*Okubo et al., 2020*). This representation is derived from a set of ring (R1) neurons that carry wind information from the LAL—silencing R1 neurons abolishes most wind responses in E-PGs (*Okubo et al., 2020*). Other sets of ring neurons also carry visual signals to E-PGs (*Omoto et al., 2017*; *Fisher et al., 2019*; *Turner-Evans et al., 2020*). Thus, in the EB, ring neurons appear to carry landmark signals of diverse modalities that collectively anchor the E-PG heading representation, while P-ENs provide angular velocity signals that rotate this representation in the absence of landmark cues (*Green et al., 2017*; *Turner-Evans et al., 2017*; *Turner-Evans et al., 2020*). This model is consistent with the fact that we did not observe strong sensory responses to airflow in P-ENs, although they are prominent in E-PGs (*Okubo et al., 2020*).

In contrast to the E-PG airflow 'compass', we found that ventral P-FNs represent airflow as a set of basis vectors tuned to two orthogonal directions, each originating ~45° to the left or right of the fly midline. Based on our imaging data, we think this representation is most likely inherited from LNa neurons, which similarly represent airflow along two orthogonal axes. This organization of airflow information strongly resembles that of optic flow signals in bee TN1 neurons, which similarly connect the LAL to the NO and provide input to PFN-type neurons (*Stone et al., 2017*). Two recent studies also identified a basis vector representation of proprioceptive and visual self-motion cues in the two dorsal PFN cell types—P-F$_3$N$_{2d}$ and P-F$_3$N$_{2v}$, also known as PFNd and PFNv (*Lu et al., 2020*; *Lyu et al., 2020*). Thus, the LAL-to-PFN system may represent flow and movement information from various modalities, organized as sets of orthogonal basis vectors. Intriguingly, the compass-like and basis set representations of airflow in the EB and FB are likely to arise from a common input pathway in the LAL (*Figure 8*). This branching of airflow information may reflect the fly's need to consider sensory signals in both allocentric (compass) and egocentric (basis vector) reference frames.

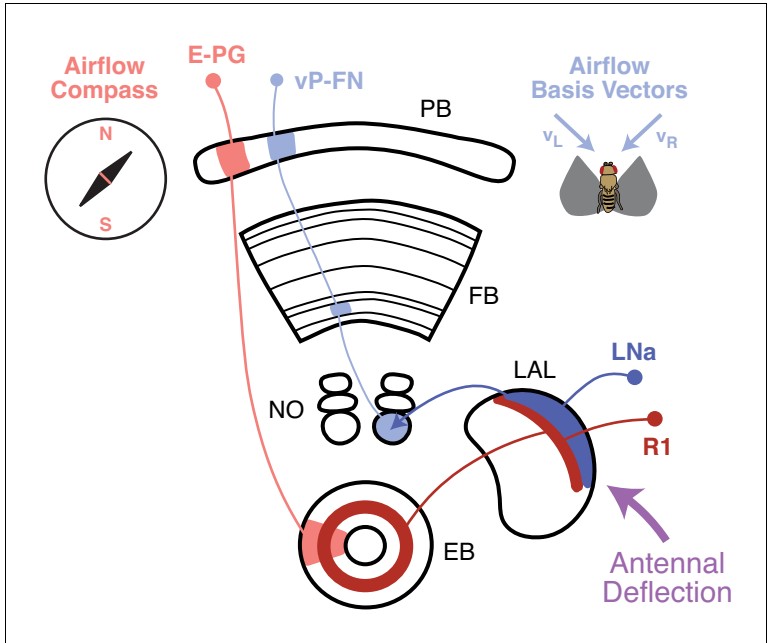

**Figure 8.** Airflow-representing circuits in the CX. Airflow direction is transduced via antennal deflection signals (purple), which are transmitted through the AMMC and Wedge to the LAL (*Yorozu et al., 2009*; *Suver et al., 2019*; *Okubo et al., 2020*). A recent study *Okubo et al., 2020* found that bilateral antennal deflection signals in the LAL are transmitted to the EB via ring neurons (R1, dark red). R1 neurons are required for formation of an airflow 'compass' in E-PG neurons (light red), which represent all possible airflow directions across CX columns. In this study, we showed that LNa neurons (dark blue, one per side) are preferentially excited by ipsilateral airflow. LNa neurons carry airflow information from the LAL to the third comparment of the ipsilateral NO (*Wolff and Rubin, 2018*). Ventral P-FNs with cell bodies in one hemisphere each receive input in the third compartment of the contralateral NO. Consistent with this anatomy, ventral P-FNs (light blue) represent airflow from only two directions (appx. 45˚ to the left and right of midline) across CX columns. Thus, airflow information encoded in the LAL appears to be routed to two different parts of the CX, where it contributes both to a heading compass representation in the EB, and a basis vector representation in the FB.

## A role for the FB in sensory orientation

Although the CX is broadly required for complex sensory navigation, the precise behavioral role of different CX compartments is still not clear. *Martin et al., 2015* found that electrode stimulation at many locations in the FB can produce reliable walking trajectories, suggesting a fairly direct role in locomotor action selection. In contrast, silencing of EB compass neurons (E-PGs) does not impair basic sensory orienting to a visual landmark, but only orienting at a fixed offset (*Giraldo et al., 2018*; *Green et al., 2019*). These studies support a model in which different CX compartments, such as the EB and FB, support different aspects of navigation behavior.

Here we found that silencing two classes of ventral P-FNs, but not E-PGs, impaired stable orientation to airflow. When we silenced ventral P-FNs, flies still turned toward the airflow source at flow off, suggesting that these cells are not required to detect airflow or to determine its direction. Flies with silenced ventral P-FNs also exhibited a normal range of motor behavior, arguing that ventral P-FNs do not generate the pool of possible responses to changing airflow direction. Instead, our data suggest that ventral P-FNs guide selection from this pool, specifically converting mechanically detected changes in orientation into an appropriate turning response. These results are broadly consistent with the idea that the FB participates more directly in basic sensory orienting (*Honkanen et al., 2019*). A caveat is that our strongest effects were observed using a broad line (44B10-GAL4) that labels both classes of ventral P-FNs, as well as some other cells in the CX and other regions. Thus, it remains possible that the more striking phenotype observed in this line arose from off-target expression. However, qualitatively similar effects were observed when only one class

of ventral P-FN was silenced. We did attempt to silence several other lines that broadly label ventral P-FNs, but we were unable to find such a line that was viable. Experiments using temporally restricted silencing may help resolve this issue.

### The role of ventral P-FNs in natural behavior

Although we have shown that silencing ventral P-FNs impairs airflow orientation in a tethered flight paradigm, the role they might play in free flight is unclear. During flight, a steady-state wind does not displace mechanoreceptors, but rather displaces the fly (*Reynolds et al., 2010*; *Leitch et al., 2020*), generating a strong optic flow signal (*Mronz and Lehmann, 2008*; *Theobald et al., 2010*) that is often offset from a fly's heading direction. However, gusts or sudden changes in wind direction can transiently activate antennal mechanoreceptors in flight, leading to behavioral responses (*Fuller et al., 2014*). Knowing the true direction of the wind in flight would be useful to the fly, both to control dispersal (*Leitch et al., 2020*), and to orient upwind toward an odor source in flight (*van Breugel and Dickinson, 2014*). Because ventral P-FNs receive input in the protocerebral bridge (PB), presumably carrying heading information from the compass system (*Franconville et al., 2018*), they may be well-poised to perform this computation. Alternatively, ventral P-FNs might be involved in estimating the direction or speed of self-motion for the purposes of course-control or memory formation (*Stone et al., 2017*). Future experiments investigating the interaction of airflow, optic flow, and heading signals in these neurons in closed loop, as well as experiments silencing these neurons during free flight, will provide insight into their function during more natural behaviors.

A final question is how ventral P-FNs are able to control steering to influence orientation to airflow. A small number of descending neurons (DNs) that participate in control of steering during flight have been identified (*Schnell et al., 2017*), although a larger number of DNs target wing motor regions and presumably play a role in flight control (*Namiki et al., 2018*). The pathways connecting the CX to these DNs have not yet been elucidated. Ventral P-FNs make most of their outputs in the FB, where they synapse onto a large number of FB local neurons and FB output neurons (*Scheffer et al., 2020*; *Hulse et al., 2020*). Many of these FB local neurons also receive input from tangential FB inputs, which may carry varied sensory signals. This arrangement could allow sensory inputs such as odor to influence orientation to airflow (*Álvarez-Salvado et al., 2018*; *van Breugel and Dickinson, 2014*), or for wind cues to be ignored if stronger visual cues are present (*Müller and Wehner, 2007*; *Dacke et al., 2019*). Future work aimed at identifying CX output pathways will be critical for understanding how ventral P-FNs and other CX neurons influence ongoing locomotor activity.

# Materials and methods

**Key resources table**

| Reagent type (species) or resource | Designation | Source or reference | Identifiers | Additional information |
|---|---|---|---|---|
| Genetic reagent (*D. melanogaster*) | SS52244-GAL4 | Bloomington *Drosophila* Stock Center | RRID:BDSC_86596 | |
| Genetic reagent (*D. melanogaster*) | SS02255-GAL4 | Bloomington *Drosophila* Stock Center | RRID:BDSC_75923 | |
| Genetic reagent (*D. melanogaster*) | SS00078-GAL4 | Bloomington *Drosophila* Stock Center | RRID:BDSC_75854 | |
| Genetic reagent (*D. melanogaster*) | SS52577-GAL4 | Bloomington *Drosophila* Stock Center | RRID:BDSC_86625 | |
| Genetic reagent (*D. melanogaster*) | SS54295-GAL4 | Bloomington *Drosophila* Stock Center | RRID:BDSC_86624 | |

*Continued on next page*

*Continued*

| Reagent type (species) or resource | Designation | Source or reference | Identifiers | Additional information |
|---|---|---|---|---|
| Genetic reagent (*D. melanogaster*) | *SS02239-GAL4* | Bloomington *Drosophila* Stock Center | RRID:BDSC_75926 | |
| Genetic reagent (*D. melanogaster*) | *SS54549-GAL4* | Bloomington *Drosophila* Stock Center | RRID:BDSC_86603 | |
| Genetic reagent (*D. melanogaster*) | *SS47432-GAL4* | Bloomington *Drosophila* Stock Center | RRID:BDSC_86716 | |
| Genetic reagent (*D. melanogaster*) | *R12D09-GAL4* | Bloomington *Drosophila* Stock Center | RRID:BDSC_48503 | |
| Genetic reagent (*D. melanogaster*) | *R44B10-GAL4* | Bloomington *Drosophila* Stock Center | RRID:BDSC_50202 | |
| Genetic reagent (*D. melanogaster*) | *(empty)-GAL4* | Bloomington *Drosophila* Stock Center | RRID:BDSC_68384 | |
| Genetic reagent (*D. melanogaster*) | *10xUAS-IVS-syn21-GFP-p10 (attP2)* | Michael Dickinson | N/A | |
| Genetic reagent (*D. melanogaster*) | *13xUAS-Kir2.1-eGFP/TM3* | Michael Reiser | N/A | |
| Genetic reagent (*D. melanogaster*) | *20xUAS-GCaMP6f* | Bloomington *Drosophila* Stock Center | RRID:BDSC_42747 | |
| Genetic reagent (*D. melanogaster*) | *UAS-tdTomato* | Bloomington *Drosophila* Stock Center | RRID:BDSC_36328 | |
| Antibody | (mouse monoclonal) anti-NC82 | Developmental Studies Hybridoma Bank | RRID:AB_2314866 | (1:50) |
| Antibody | (chicken polyclonal) anti-GFP | Thermo Fisher Scientific | PA1-9533 | (1:50) |
| Antibody | streptavidin Alexa Fluor 568 | Thermo Fisher Scientific | S-11226 | (1:1000) |
| Antibody | (goat polyclonal) anti-mouse Alexa Fluor 633 | Thermo Fisher Scientific | A-21052 | (1:250) |
| Antibody | (goat polyclonal) anti-chicken Alexa Fluor 488 | Thermo Fisher Scientific | A-11039 | (1:250) |

## Contact for reagent and resource sharing

Further information and requests for resources and reagents should be directed to and will be fulfilled by the Lead Contact, Katherine Nagel (katherine.nagel@nyumc.org).

## Experimental model and subject details

### Fly stocks

All flies were raised at 25°C on a cornmeal-agar medium under a 12-hr light/dark cycle. Flies for patch experiments were aged 1–3 days after eclosion before data collection, and flies for behavior experiments were aged 3–5 days. All data shown are from female flies. Parental stocks can be found in the key resources table. Specific genotypes presented in each figure panel are shown below. SS

lines contain genetic inserts on chromosomes II and III—the transgenes occupying the second copies of these chromosomes are shown in parenthesis.

| Genotype | N | Description | Figure panels |
|---|---|---|---|
| w; SS52244-GAL4/(+; 10xUAS-syn21-GFP) | 6 | P-$F_1N_3$ patch line | 1D,E,G,H |
| w; SS02255-GAL4/(+; 10xUAS-syn21-GFP) | 6 | P-$F_2N_3$ patch line | 1C,E,G,H |
| w; SS00078-GAL4/(+; 10xUAS-syn21-GFP) | 14 | P-$F_3N_{2d}$ patch line | 1E–H |
| w; SS52577-GAL4/(+; 10xUAS-syn21-GFP) | 4 | P-$F_3N_{2v}$ patch line | 1E,G,H |
| w; SS54295-GAL4/(+; 10xUAS-syn21-GFP) | 4 | P-$EN_1$(1) patch line | 1E,G,H |
| w; +; R12D09-GAL4/10xUAS-syn21-GFP | 6 | P-$EN_1$(2) patch line | 1C,E,G,H |
| w; SS02239-GAL4/(+; 10xUAS-syn21-GFP) | 8 | P-$F_3$LC patch line | 1C,E,G,H |
| w; SS54549-GAL4/(+; 10xUAS-syn21-GFP) | 6 | P-$F_{3-5}$R patch line | 1E,G,H |
| w; SS52244-GAL4/(+; 10xUAS-syn21-GFP) | 6 | P-$F_1N_3$ patch line | 2A |
| w; SS02255-GAL4/(+; 10xUAS-syn21-GFP) | 6 | P-$F_2N_3$ patch line | 2A |
| w; SS00078-GAL4/(+; 10xUAS-syn21-GFP) | 14 | P-$F_3N_{2d}$ patch line | 2A,B–E |
| w; SS52577-GAL4/(+; 10xUAS-syn21-GFP) | 4 | P-$F_3N_{2v}$ patch line | 2A |
| w; SS54295-GAL4/(+; 10xUAS-syn21-GFP) | 4 | P-$EN_1$(1) patch line | 2A |
| w; +; R12D09-GAL4/10xUAS-syn21-GFP | 5 | P-$EN_1$(2) patch line | 2A,B–E |
| w; SS02239-GAL4/(+; 10xUAS-syn21-GFP) | 8 | P-$F_3$LC patch line | 2A |
| w; SS54549-GAL4/(+; 10xUAS-syn21-GFP) | 6 | P-$F_{3-5}$R patch line | 2A |
| w; SS02255-GAL4/(+; 10xUAS-syn21-GFP) | 6 | P-$F_2N_3$ patch line | 3A–G |
| w; SS52244-GAL4/(+; 10xUAS-syn21-GFP) | 6 | P-$F_1N_3$ patch line | 3H,I |
| w; SS02255-GAL4/(+; 10xUAS-syn21-GFP) | 12 | P-$F_2N_3$ patch line | 4B-K |
| w; SS02255-GAL4/(+; 10xUAS-syn21-GFP) | 6 | P-$F_2N_3$ patch line | 5C |
| w; +; VT029515-GAL4/10xUAS-syn21-GFP | 8 | vFBN patch lines | 5C |
| w; SS47432-GAL4/(20XUAS-GCaMP6f; UAS-tdTomato) | 5 | LNa imaging line | 5B,C |
| w; +; R44B10-GAL4/13xUAS-Kir2.1-eGFP | 12 | Ventral P-FN silencing line | 6B–F |
| w; +; (empty)-GAL4/13xUAS-Kir2.1-eGFP | 12 | Silencing control line | 6 C–F |
| w; SS02255-GAL4/(+; 13xUAS-Kir2.1-eGFP) | 11 | P-$F_2N_3$ silencing line | 6 C–F |
| w; SS52244-GAL4/(+; 13xUAS-Kir.21-eGFP) | 11 | P-$F_1N_3$ silencing line | 6 C–F |
| w; SS00090-GAL4/(+; 13xUAS-Kir.21-eGFP) | 11 | E-PG silencing line | 6 C–F |
| w; +; R44B10-GAL4/13xUAS-Kir2.1-eGFP | 12 | Ventral P-FN silencing line | 7B–G |
| w; +; (empty)-GAL4/13xUAS-Kir2.1-eGFP | 12 | Silencing control line | 7B–G |
| w; SS02255-GAL4/(+; 13xUAS-Kir2.1-eGFP) | 11 | P-$F_2N_3$ silencing line | 7C–E,G |
| w; SS52244-GAL4/(+; 13xUAS-Kir.21-eGFP) | 11 | P-$F_1N_3$ silencing line | 7C–E,G |
| w; SS00090-GAL4/(+; 13xUAS-Kir.21-eGFP) | 11 | E-PG silencing line | 7C–E,G |

## Method details

### Electrophysiology

Flies were prepared for electrophysiology (*Figures 1–4*) by tethering them to a custom fly holder and reservoir (modified from *Weir and Dickinson, 2015,*). All flies were cold anesthetized on ice for approximately 5 min during tethering. During anesthesia, we removed the front pair of legs from each fly to prevent disruption of electrophysiology data. We then used UV glue (KOA 30, Kemxert) to fix flies to the holder by the posterior surface of the head and the anterior-dorsal thorax. Once flies were secured, they were allowed to recover from anesthesia for 30–45 min in a humidified chamber at 25°C. Prior to patching, we filled the fly holder reservoir with *Drosophila* saline (*Wilson and Laurent, 2005*, see below) and dissected away the cuticle on the posterior surface of the head, removing trachea and fat lying over the posterior surface of the brain.

Tethered and dissected flies were then placed in a custom stimulus arena on a floating table (Technical Manufacturing Corporation, 63–541) with a continuous flow of room-temperature *Drosophila* saline over the exposed brain. Briefly, *Drosophila* saline contained 103 mM sodium chloride, 3 mM potassium chloride, 5 mM TES, 8 mM trehalose dihydrate, 10 mM glucose, 26 mM sodium bicarbonate, 1 mM sodium phosphate monohydrate, 1.5 mM calcium chloride dihydrate, and 4 mM magnesium chloride hexahydrate. The solution was adjusted to a pH of 7.2 and an osmolarity of 272 mOsm.

Brains were imaged under 40X magnification (Olympus, LUMPLFLN40XW) by a microscope (Sutter, FG-SOM-XT) controlled by a micromanipulator (Sutter, MPC-200). Real-time brain images were captured by a camera (Dage-MTI, IR-1000) and sent to an LCD monitor (Samsung, SMT-1734). Target neurons were identified based on expression of cytoplasmic GFP (see 'Fly Stocks' above). Fluorescent stimulation was provided by an LED light source and power controller (Cairn Research, MONOLED). A dichroic/filter cube (Semrock, M341523) allowed for stimulation and emission imaging through the same objective.

We first cleared away the neural sheath overlaying target neurons by puffing 0.5% collagenase-IV (Worthington, 43E14252) through a micropipette (World Precision Instruments, TW150-3) and applying gentle mechanical force. Cell bodies overlaying target somata were then removed via gentle suction, if necessary. Once target neurons were cleaned of debris, we used fire-polished micropipettes (*Goodman and Lockery, 2000*) to record one neuron per fly. Prior to use, pipettes (World Precision Instruments, 1B150F-3) were pulled (Sutter, Model P-1000 Micropipette Puller) and transferred to a polishing station equipped with an inverted light microscope and a pressurized micro-forge (Scientific Instruments, CPM-2). Pipettes were polished to a tip diameter of 0.5–2 µm and an impedance of 6–12 MΩ, depending on the target cell type. Patch pipettes were filled with potassium-aspartate intracellular solution (*Wilson and Laurent, 2005*), which contained 140 mM of potassium hydroxide, 140 mM of aspartic acid, 10 mM of HEPES, 1 mM of EGTA, 1 mM of potassium chloride, 4 mM of magnesium adenosine triphosphate, and 0.5 mM of trisodium guanine triphosphate. We also added 13 mM of biocytin hydrazide to the intracellular solution for post-hoc labeling of recorded neurons. The solution was adjusted to a pH of 7.2 and an osmolarity of 265 mOsm. Before use, we filtered this intracellular solution with a syringe-tip filter (0.22 micron pore size, Millipore Millex-GV).

Recorded neurons were confirmed to be of the targeted type in three ways: (1) presence of a fluorescent membrane 'bleb' inside the pipette after sealing onto the cell; (2) loss of cytoplasmic GFP through diffusion over the course of the recording session; (3) post-hoc biocytin fill label matching the known anatomical features of the targeted cell type. Two of these three criteria must have been met in order for a patched neuron to be considered a member of a given cell type.

Hardware for electrophysiology was adapted from one previously used (*Nagel and Wilson, 2016*). In brief, recorded signals passed through a headstage (Axon Instruments, CV 203BU), an amplifier (Molecular Devices, Axopatch 200B), and a pre-amp (Brownlee Precision, Model 410), before being digitized for storage (National Instruments, PCIe-6321) and gain-corrected. Data was collected at 10,000 Hz.

After all electrophysiology experiments, we removed flies from their holders and dissected out their central brains. Dissected tissue was then fixed in 4% paraformaldehyde for 14 min at room temperature. Fixed tissue was stored at 4˚C for up to 4 weeks before further processing (see 'Immunohistochemistry,' below).

## Stimulus delivery

Once an active recording was obtained, we presented a series of sensory stimuli from multiple directions. Stimulus delivery was achieved by a modified version of a previously used system (*Currier and Nagel, 2018*). Briefly, custom LabView (National Instruments) software controlled the triggering of airflow (25 cm/s), odor (20% apple cider vinegar), and/or ambient illumination (15 µW/cm$^2$) of a high contrast vertical bar that subtended approximately 30˚ of visual angle. Stimulus intensities for airflow, odor, and light were measured with a hot-wire anemometer (Dantec Dynamics MiniCTA 54T42), a photo-ionization detector (Aurora Scientific, miniPID 200B), and a power meter (ThorLabs, PM 100D and S130C), respectively. All stimuli emanated from the same location in the arena, and the entire arena could be rotated with a stepper motor (Oriental Motor, CVK564FMBK) around the stationary

fly. This setup allowed us to present cues from any arbitrary direction. Rotations of the motor were slow (20°/s) and driven at minimal power to minimize vibration and electromagnetic disturbances.

For our initial survey of CX columnar neurons (*Figures 1–3*), we used a pseudorandom session design broken down by stimulus direction (−90˚, 0˚, 90˚, and 180˚) and type (stripe only, airflow only, airflow and stripe together, airflow and odor together, or all three stimuli simultaneously). Each 12 s trial included 4 s of pre-stimulus baseline, 4 s of stimulus presentation, and 4 s of post-stimulus time. The first 1 s of each trial's baseline period included a 500 ms injection of −2 pA to monitor input resistance over time (this period is not plotted in any Figures). Between trials, a 9 s inter-trial-interval allowed the cell to rest while the motor rotated the arena to the next trial's stimulus direction. All 20 unique combinations of stimulus direction and condition were presented four times each, and each stimulus was presented before the next round of repetitions began. The total session recording time was approximately 50 min. If cell health was observed to decay before the session was complete, data were discarded after the preceding 'set' of 20 stimuli. For a cell to be included in the survey, at least 40 trials (2 sets of repetitions) must have been completed. Of the 52 neurons patched in the survey, 46 remained healthy for all 80 trials.

To investigate how airflow responses varied by column (*Figure 4*), we used eight directions (−135˚, −90˚, −45˚, 0˚, 45˚, 90˚, 135˚, 180˚) and presented only the airflow stimulus. Trial and pseudo-random session design were the same as above, but we increased the number of stimulus repetitions to 5, for a total of 40 trials. All 12 flies in this dataset completed all 40 trials.

## Immunohistochemistry

Fixed brains were processed using standard immunohistochemistry protocols. Briefly, we blocked for 30 min at room temperature in phosphate buffered saline (PBS, Sigma, P5493-1L) containing 5% normal goat serum (Vector Laboratories, S-1000) and 0.1% Triton X-100 (Sigma, X100-100ML). The primary antibody solution was identical to the blocking solution, but had a 1:50 dilution of chicken anti-GFP antibodies (Fisher Scientific, A-6455) and a 1:50 dilution of mouse anti-bruchpilot antibodies (Developmental Studies Hybridoma Bank, nc82-s). The secondary antibody solution was similarly based on the blocking solution, but also contained a 1:250 dilution of alexa488-conjugated goat anti-chicken anitbodies (Fisher Scientific, A-11034), a 1:250 dilution of alex633-conjugated goat anti-mouse antibodies (Fisher Scientific, A-21052), and a 1:1000 dilution of alexa568-conjugated streptavidin (Fisher Scientific, S-11226). Antibody incubations were for 24 hr at room temperature. We washed brains in 0.1% PBS-Triton three times for 5 min after each antibody phase. Immuno-processed brains were mounted on slides (Fisher Scientific, 12-550-143 and 12–452˚C) and imaged under a confocal fluorescence microscope (Zeiss, LSM 800) at 20X magnification (Zeiss, W Plan-Apochromat 20x).

## Calcium imaging

For calcium imaging, flies (age 10 to 16 days) were anesthetized and mounted in a simpler version of our electrophysiology holder. Flies were starved for 18–24 hr prior to beginning the experiment. The back cuticle of the head was dissected away using fine forceps and UV glue was applied to the fly's proboscis to prevent additional brain movement. Flies were allowed to recover for 5 min prior to imaging. The holder chamber was filled with *Drosophila* saline (as above) and perfused for the duration of imaging.

2-photon imaging was performed using a pulsed infrared laser (Mai Tai DeepSea, SpectraPhysics) with a Bergamo II microscope (Thorlabs) using a 20x water-immersion objective (Olympus XLUMPLFLN 20x) and ThorImage 3.0 software. Laser wavelength was set to 920 nm and power at the sample ranged from 13 to 66 mW. Emitted photons were spectrally separated using two band-pass filters (red, tdTOM: 607/70 nm, green, GCaMP: 525/50 nm) and detected by GaAsP PMTs. The imaging area of approximately $132 \times 62$ μM was identified using the tdTOM signal. Imaging was performed at 5.0 frames per second.

Airflow and odor stimuli were delivered using a fixed 5-direction manifold (*Suver et al., 2019*) and controlled by proportional valves (EVP series, EV-05–0905; Clippard Instrument Laboratory, Inc Cincinnati, OH) using custom Matlab code running on its own PC. We used a hot-wire anemometer (Dantec Dynamics MiniCTA 54T42) to verify that airspeed (~30 cm/s) was equivalent from all five directions and did not change during odor delivery. Odorant (apple cider vinegar) was diluted to

1:10 in distilled water on the day of the experiment. Stimuli consisted of 10 s of airflow, 10 s of airflow plus odor, followed by another 10 s of airflow with 5 s before and 12 s after stimulus presentation. The order of airflow direction was randomized in each block of five trials and we performed five blocks per fly. After each block the imaging frame was re-adjusted to account for any drift, and gain and power level were optimized. One fly was excluded from the final analysis as it failed to respond to any stimuli. All flies included include all trials from all five blocks.

### Behavior

For flight simulator experiments (*Figure 5* and *6*), we fixed flies in place using rigid tungsten tethers (see *Currier and Nagel, 2018*). All flies were cold anesthetized for approximately 5 min during the tethering process. During anesthesia, a drop of UV-cured glue was used to tether the notum of anesthetized flies to the end of a tungsten pin (A-M Systems, # 716000). Tethered flies' heads were therefore free to move. We additionally removed the front pair of legs from each fly to prevent disruption of wing tracking (see below). Flies were then allowed to recover from anesthesia for 30–60 min in a humidified chamber at 25℃ before behavioral testing.

Tethered behavior flies were placed one at a time in a custom stimulus arena described in a previous paper (*Currier and Nagel, 2018*). The dark arena was equipped with a pair of tubes (one flow, one suction) that could create a constant stream of airflow over the fly. A camera (Allied Vision, GPF031B) equipped with a zoom lens (Edmund Optics, 59–805) and infrared filter (Midwest Optical, BP805-22.5) was used to capture images of the fly in real-time. Custom LabView software was used to detect the angle of the leading edge of each wing. We multiplied the difference between these wing angles (ΔWBA) by a static, empirically verified gain (0.04, see *Currier and Nagel, 2018*) to determine each fly's intended momentary angular velocity. This signal was sent to a stepper motor (Oriental Motor, CVK564FMAK) which rotated the airflow tubes around the fly. The difference in wingbeat angles and integrated heading were also saved for later analysis. This process was repeated at 50 Hz.

Each fly's 20 min behavioral testing session was broken down into 20 s trials that began with a manipulation to the airflow stimulus. Air flowed continuously throughout the entire session, except when interrupted as a stimulus manipulation. We used two durations of airflow pause: two samples (100 ms) and 100 samples (2 s). Additional manipulations included open-loop (not fly-controlled) rotations of the airflow tubes to the left or right of its current position. These open-loop rotations were driven continuously at the maximum speed of the motor (144 °/s) for either five samples (14.4°) or 22 samples (63.36°). This gave four additional manipulations: long and short rotations to the left and right. These six stimulus manipulations were pseudorandomly presented ten times each. All flies shown in *Figure 5* and *6* completed all 60 trials.

## Quantification and statistical analysis

### Analysis of physiology data

All data were processed in MATLAB (Mathworks, version 2017B) with custom analysis scripts. Spike times were found by first high-pass filtering the raw membrane potential signal with a second order Butterworth filter (40 Hz cutoff frequency) and then identifying cell type-specific threshold crossings. We used the −2 pA test pulse at the beginning of each trial to calculate and track input resistance over time (*Figure 1—figure supplement 1D*).

To calculate membrane potential and spiking responses to our stimuli, we first defined the baseline period for each trial as a 1 s long data segment ending 500 ms before stimulus onset. We additionally defined a response period, which began 500 ms after stimulus onset and similarly lasted 1 s. Spiking and membrane potential responses in each trial were defined as the mean firing rate or membrane potential during the response period minus the mean of the baseline period. Mean responses (*Figures 1–4* and **S2**) for each cell to each stimulus were calculated by averaging these responses to stimulus repetitions.

Cross-condition correlation coefficients (*Figures 2* and *3* and **S4**) were found by first taking the mean PSTH across stimulus repeats in each stimulus condition. PSTHs were found by convolving spike trains with a 1 s Hanning window. We next truncated this full trial mean data to only include the stimulus response and offset periods (seconds 4–12 of each 12 s trial). Truncated mean response timecourses for the four stimulus directions were then concatenated for each sensory condition. We

then found the correlation coefficient between these direction-concatenated mean PSTHs for multisensory trials (airflow + stripe) versus airflow only or stripe only.

Response-by column analyses included additional metrics. To calculate the mean orientation tuning of each neuron, we first converted the mean spiking response to each airflow direction into a vector, with an angle corresponding to the airflow direction and a magnitude equal to the mean response to that direction. We then calculated the mean vector across the response vectors corresponding to the eight stimulus directions that we used. The angle of this mean vector is plotted in *Figure 4E*.

Airflow response dynamics were examined using the response to ipsilateral airflow. We computed the cumulative sum of the PSTH during the full 4 s stimulus period, and divided this timecourse by the integral of the PSTH over the entire 4 s (*Figure 4J*). We additionally found the time when each cell's cumulative normalized response reached 0.5, or half its total response (*Figure 4K*).

## Analysis of calcium imaging data

Analysis was performed using the CalmAn Matlab package, Image J, and custom Matlab scripts. We used the CalmAn package (*Giovannucci et al., 2019*) to implement the NoRMCorre rigid motion correction algorithm (*Pnevmatikakis and Giovannucci, 2017*) on the red (tdTOM) time series and applied the same shifts to the green (GCaMP6f) time series. Regions of interest (ROIs) were drawn by hand around the left and right projections between the LAL and NO in ImageJ, using the maximum intensity projection of the tdTOM time series. ROIs were applied to all trials. The positioning of ROIs was adjusted by hand using ImageJ on any trials where significant drift occurred. ImageJ ROIs were imported into Matlab using ReadImageJROI (*Muir and Kampa, 2014*). We calculated ΔF/F for both tdTOM and GCaMP6f time series by dividing the time series by the average fluorescence of the baseline period (first 5 s of the trial). We subtracted the change in red from the change in green to correct for any fluctuations in the Z plane that occurred due to brain movement. The ΔF/F in plots refers to $\Delta F/F_{Green}$-$\Delta F/F_{Red}$.

## Analysis of behavioral data

Flight simulator data was collected at 50 Hz as the difference in wingbeat angles (ΔWBA, raw behavior – see *Figure 7*) and integrated orientation (cumulative sum of the feedback signal to the arena motor, see above). We parsed the integrated orientation data in three ways. First, we converted the data points into vectors with an angle equal to the fly's virtual orientation on that sample, and a length of 1. The mean orientation vector was then calculated for each fly. Second, we used the length of a this mean orientation vector to evaluate orientation stability. Third, we found the fraction of samples in the integrated orientation distribution that fell between −45° and 45° (the 'toward the airflow' arena quadrant). These measures are all plotted in *Figure 6*.

To assess flies' responses to our airflow manipulations, we analyzed ΔWBA during a 6 s period surrounding the stimulus manipulation (1 s pre-stimulus and 5 s post-stimulus). We found each fly's mean ΔWBA response across 10 repeats of each slip stimulus. We then found the cross-fly mean and SEM (*Figure 7F*).

We additionally integrated ΔWBA over the 5 s following each slip stimulus to calculate an angular 'response' to that slip (*Figure 7E,G,H*). Slip response magnitudes were placed into 20° bins and collapsed across slip direction for the purposes of plotting response distributions (*Figure 7E*). To calculate the fraction of each slip for which flies corrected, we divided integrated slip response on each trial by the negative magnitude of the stimulus slip, then took the mean correction fraction across all trials of a given slip magnitude (long or short, see *Figure 7G*).

To assess flies' responses to airflow pause, we modified the sign of the raw ΔWBA signal. Because airflow generally drives 'with the flow' orienting in rigidly tethered flies (*Currier and Nagel, 2018*), we wanted a direction-invariant measure of with-/away from airflow turning. Accordingly, we altered the sign of ΔWBA on each sample such that turns toward the airflow source were positive, and turns away from the airflow source were negative (normally, positive ΔWBA indicates rightward turns, and leftward for negative ΔWBA). We then calculated cross-fly mean airflow pause responses (*Figure 7B*) as described above for the slip stimuli. For long airflow pauses, we additionally integrated ΔWBA over the 2 s stimulus period (*Figure 7C*).

## Statistical analysis

All statistical analyses used non-parametric tests corrected for multiple comparisons (Bonferroni method). For single fly data, paired comparisons were made using the Wilcoxon Sign-Rank test (MATLAB function signrank), and unpaired comparisons using the Mann-Whitney U test (MATLAB function ranksum). Cross-fly (per-genotype) distributions were compared using the Kolmogorov-Smirnov test (MATLAB function kstest2). Significance values for all tests are reported in the Figure Legends.

## Connectomic analysis

Data from the fly hemibrain connectome (*Scheffer et al., 2020*) was visualized using neuPRINT explorer (neuprint.janelia.org). We identified $P\text{-}F_2N_3$ as PFNa in this dataset on the basis of NO innervation and nomenclature of *Wolff et al., 2015*. Similarly, $P\text{-}F_1N_3$ was identified as PFNm and PFNp. Based on the connectivity data for an example LNa neuron (1508956088), we determined that $P\text{-}F_2N_3$ neurons across CX columns receive significant input from LNa neurons. $P\text{-}F_2N_3$ neurons specifically receive input from contralateral LNa neurons in the NO.

# Acknowledgements

We would like to thank Michael Reiser for flies and Michael Long, Gaby Maimon, David Schoppik, and members of the Nagel and Schoppik labs for feedback and helpful discussion. This work was supported by grants from the NIH (R01DC017979 and R01MH109690), and NSF (IOS-1555933) to KIN, a McKnight Scholar Award to KIN, and a New York University Dean's Fellowship to TAC.

# Additional information

### Funding

| Funder | Grant reference number | Author |
| --- | --- | --- |
| National Institute on Deafness and Other Communication Disorders | R01DC017979 | Katherine I Nagel |
| National Institute of Mental Health | R01MH109690 | Katherine I Nagel |
| McKnight Endowment Fund for Neuroscience | Scholar Award | Katherine I Nagel |
| National Science Foundation | IOS-1555933 | Katherine I Nagel |
| New York University | Dean's Fellowship | Timothy A Currier |
| National Science Foundation | Neuronex: 2014217 | Katherine I Nagel |

The funders had no role in study design, data collection and interpretation, or the decision to submit the work for publication.

### Author contributions

Timothy A Currier, Conceptualization, Software, Formal analysis, Funding acquisition, Validation, Investigation, Methodology, Writing - original draft, Writing - review and editing; Andrew MM Matheson, Software, Formal analysis, Validation, Investigation, Writing - review and editing; Katherine I Nagel, Conceptualization, Supervision, Funding acquisition, Methodology, Writing - original draft, Writing - review and editing

### Author ORCIDs

Timothy A Currier http://orcid.org/0000-0002-1092-5563
Andrew MM Matheson http://orcid.org/0000-0001-9062-2521
Katherine I Nagel https://orcid.org/0000-0002-6701-3901

Decision letter and Author response
Decision letter https://doi.org/10.7554/eLife.61510.sa1
Author response https://doi.org/10.7554/eLife.61510.sa2

## Additional files

### Supplementary files
- Transparent reporting form

### Data availability
All electrophysiology, behavior, and anatomy data are publicly available on Dryad at https://doi.org/10.5061/dryad.vq83bk3rh.

The following dataset was generated:

| Author(s) | Year | Dataset title | Dataset URL | Database and Identifier |
|---|---|---|---|---|
| Currier TA, Matheson AMM, Nagel KI | 2020 | Encoding and control of airflow orientation by a set of Drosophila fan-shaped body neurons | https://doi.org/10.5061/dryad.vq83bk3rh | Dryad Digital Repository, 10.5061/dryad.vq83bk3rh |

The following previously published datasets were used:

| Author(s) | Year | Dataset title | Dataset URL | Database and Identifier |
|---|---|---|---|---|
| Xu CS | 2020 | A connectome of the Adult Drosophila Central Brain | https://neuprint-test.janelia.org/?dataset=hemibrain&qt=findneurons | Neuprint, 10.1101/2020.01.21.911859 |
| Clements J | 2020 | neuPrint: Analysis Tools for EM Connectomics | https://neuprint-test.janelia.org/?dataset=hemibrain&qt=findneurons | Neuprint, 10.1101/2020.01.16.909465 |

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
