## [Decision Letter]

**Acceptance summary:**

Currier et al., is an impressive and important contribution to understanding the *Drosophila* central complex (CX). The authors perform technically challenging electrophysiological recordings from many identified CX neurons, focusing on the not yet characterized P-FN neurons. They find that ventral P-FNs in each hemisphere were tuned to 45 degrees ipsilateral, forming a pair of orthogonal bases. These findings will drive the development of ideas about how other subtypes of PFN neurons might respond to sensory information, and thereby will aid the development of a unified idea of how all aspects of the sensory world perceived by the fly can influence action selection via the CX circuits.

**Decision letter after peer review:**

Thank you for submitting your article "A central complex population that supports action selection during orientation to airflow" for consideration by *eLife*. Your article has been reviewed by Ronald Calabrese as the Senior and Reviewing Editor and three reviewers. The reviewers have opted to remain anonymous.

The reviewers have discussed the reviews with one another and the Reviewing Editor has drafted this decision to help you prepare a revised submission.

Summary:

Currier et al., performed patch-clamp recordings from columnar neurons in the fly central complex. They discovered that ventral PFNs respond strongly to wind stimuli and rather than forming a map of wind directions, all P-FNs on one side of the brain respond to wind with a peak tuning of ~45° to the right of the fly and vice versa for ventral P-FNs on the other side of the brain. Silencing ventral P-FNs impairs tethered wind-orienting flight behavior while preserving basic sensory and motor capacities. This is a generally informative study with high quality data important to understanding navigation in flies and potentially in outer systems.

Essential revisions:

1) The reviewers were generally concerned because their understanding of the connectomics is that the LNs predominantly provide inputs to the PFNs and not predominantly outputs to the LAL. In reviewer consultation "…the LN neurons in question are typically considered inputs to the CX, not outputs, as posited in the authors' final model. As such, …the authors' model [is]... odd. One main reason to believe that LNs provide inputs to the PFNs is because the connectome reveals these cells to receive thousands of input synapses in the LAL and to have thousands of output synapses in the noduli, on PFNs. That said, formally, the LNs in question do have ~30 output chemical synapses in the LAL, and gap junctions are also possible. So... while far from conventional, … one could, in principle, posit an output role, for those neurons…" There was skepticism that PFNs send signals out of the CX via the LN neurons, and if the authors adhere to it in revision they must make every attempt to rationalize this conception, discuss the # of synapse issue, and minimally introduce other, more conventional, models for how control of behavior by these neurons could work, via intermediaries in the fan-shaped body.

2) The authors patch-clamped cells in more restrictive split-Gal4's for the ventral PFNs and these lines gave the weaker behavioral phenotypes. R44B10 was used to improve the rigor of the behavioral results. R44B10 has labeling outside of the CX, (it's not the worst line; the cells targeted outside of the CX number in the dozens to hundreds -- not bad for a generation-1 Gal4 line -- not thousands). Rather than perform additional experiments (as stated in the expert reviews) to ensure that the behavioral effects are mediated by ventral PFNs, the authors could perform additional time-series analyses or depictions of their behavioral genetics results with the three Gal4 lines that they used to date, to make clear that the phenotypes in the two split-Gal4 lines are indeed robust. Our concern is that three left-most columns in Figure 5D just don't look that different, and this is a standard quantification. Discussion should state very explicitly the limitations/concerns about using Gal4 lines in the behavioral experiments.

3) All three expert reviewers agree that the findings described in this study are not well integrated with previous literature, specifically in relation to wind sensing within Central Complex circuits, and in relation to processing of information in other compartments. There was also confusion on the rules underlying multisensory interactions in PFNs and the authors should make very clear that they are examining multimodal interactions in a fixed condition.

The detailed expert reviews amplify upon these three basic concerns and will guide the authors in revision.

Reviewer #1:

In this study, Currier et al., examine the sensitivity to airflow, a visual landmark and olfactory cues across a limited subset of so-called columnar neurons within the Central Complex (Cx) of the fly, a navigational center of the insect brain. They found that a subset of FB columnar cells displays robust responses to directional airflow, and that activity across the population does not follow an abstract representation of heading as was recently reported for another set of columnar neurons in the Cx. Instead, this population seem to report a signal related to a contrast between sides across the body. Finally, they show that genetic-based manipulations of the activity of these cells promote unstable orientations, while not affecting neither motor programs nor the detection of the location of the airflow source. Instead, the authors propose that an improper selection of the motor program underlies the degradation of the stabilization of the oriented response. Overall, I find that these findings are timely, and supported by the experimental evidence largely. However, I find some disconnection between the physiology and the behavior, and between their findings and previous work on airflow processing in Cx columnar cells. In addition, I think that the way the mulmodal interactions are described may be a bit misleading. The specific comments below touch on these points

Link between physiology and behavior:

The apparent distinct representation of the population of ventral P-FNs compared to that reported for P-ENs and E-PGS is an important aspect of this study. Specifically, the lack of a systematic map of airflow direction at ventral P-FNs, but rather a contrast base signal. Notably, the authors make no comment about how this representation may change from the allocentric description of airflow location, to one that incorporates egocentric information (for example, heading). I think this is an important point to emphasize the importance of their findings. I would encourage a discussion along this direction especially because they seem to connect the idea that a simple "bases" representation may be adequate for a moment-by-moment action selection. Because these populations of neurons are proposed to be closer to motor programs rather that abstract representations, an egocentric reference frame may be a more appropriate description of population dynamics. If this were the case, a conjunctive interaction is expected with an updated information about heading. In reality, focusing their analysis in the absence of movement in a head-fixed preparation essentially makes the allocentric and egocentric frames apparently overlapping. By I wonder whether by manipulating different signals, such as the visual and the airflow, and making them appear at different locations, may reveal a multimodal interaction that is link to the transformation between abstract onto a more motor reference frame, of whether the authors think this transformation may happen somewhere else, for example, at the level of DNs.

Certainly, the behavioral experiments have an internal signal related to the ongoing movements of the fly, that likely is integrated with the sensory signal. The question is whether this integration happens at the ventral layers of the FB or not.

Multimodal interactions:

If I understood correctly, all stimuli were placed at the same location. I think this may be an important point in the conclusions and interpretations for multisensory interactions. I would encourage to make more explicit the point that the apparent additive effect of multimodal processing is specifically related to the condition that the spatial information of the stimuli is congruent across modalities. It would be interested to see how these interactions change as a function of a dissonant spatial information across modalities.

Integration of the work with previous literature:

Although the authors describe and acknowledge previous work on airflow processing in Cx, it is a bit disappointing to see a lack of integration of the present findings with those reported before. Specifically, are these airflow signals also dependent on R1? Why are the authors proposing the ventral-FNs are an alternative entrance point of airflow processing? Are they coming from the antennae or bristles all over the body?, etc. Why do they think that E-PGs but not P-ENs are so robust to airflow? What is/are the expected path/s?

Reviewer #2:

The manuscript by Currier et al., is an impressive and important contribution to understanding the insect central complex. It provides highly relevant data for understanding the role of this brain structure in guiding navigation behavior based on multisensory input. Using *Drosophila* as their model, the authors perform electrophysiological recordings from many identified CX neurons, in particular the so far not characterized PFN neurons. These neurons have been recently proposed to represent the desired heading of the insect (based on data in bees) and the presented data not only corroborate the general hypotheses of how the CX circuitry might guide behavior, but it presents a wealth of detailed data on the role specific types of PFN neurons play in this context in flies. This will drive the development of ideas about how other subtypes of PFN neurons might respond to sensory information, and thereby will aid the generation of a unified idea of how all aspects of the sensory world perceived by the fly can influence action selection via the CX circuits.

The technically challenging work is very well carried out, the data is of high quality, controls are in place, and I enjoyed reading the results part a lot. I have one major issue with stimuli, which, however, is mostly relevant for future studies and will only require re-evaluating some conclusions in the present work.

There is one major issue in the paper as it is, and this refers to the discussion and the general conclusions about the role the examined neurons play in the overall CX circuitry. As explained in detail below, this appears to be based on a misinterpretation of the hemibrain data. Interestingly, if revised, the altered conclusions (taking into account the real synaptic distributions of the PFN and LN neurons) fit much more nicely with the data available from other species and push our understanding of the CX into a direction of a generally applicable hypothesis of the neural basis of insect navigation.

Essential revisions:

1) (sorry if this point is very long) The authors make use of the recently published connectome data, but they appear to have misinterpreted the polarity of their neurons, leading to conclusions that are opposite to the available data in flies, and also at odds with the data from other species. The final circuit hypothesis and the associated conclusions have to therefore be revised to a large extent.

Details: Opposite to what the authors state in the Materials and methods and in the Discussion, the lateral neurons of the noduli, the LN neurons (specifically the LNa and LCNpm) do not have outputs in the LAL and (prime) inputs in the noduli. Within the noduli these cells have only outputs (95 and 98%) and in the LAL these cells have only input synapses. It is correct that they also receive inputs from PFNs in the Noduli, but given the lack of outputs outside the Noduli, these must be local feedback connections, not influencing the overall direction of information flow.

The LN neurons hence provide input to the PFN cells and are no possible output pathway. The only output that relays information from the PFN cells to other brain regions is via the FB (as there these cells have almost only output synapses). Targets are columnar neurons that project either to the LAL (PFL1) and to the Crepine (FC1 cells, formerly FQ6 cells). This generally is in line with the information flow suggested by Stone et al., (2017), Honkanen et al., (2019) and Franconville et al., (2019). Here, the PFL cells (or related columnar neurons) are the main output pathway from the CX, combining information from the PB (for PFL cells) and the FB to project to the LAL/CRE regions. The PFN cells are suggested to input onto these output neurons, which integrate multiple sources of information (different PFNs, tangential FB cells, delta7 cells in the PB) and could provide a steering signal to the LAL (creating an imbalance between the summed right and left output).

The presented data is highly consistent with this idea and the evidence for downstream processing (e.g. DNa02 neurons) is not contradictory to this circuit either (as it receives input from PFL cells).

2) Stimuli: The tested neurons likely receive input from two sources: the head direction system (via the PB, potentially inhibitory through the delta7 cells) and an input via the noduli (see major point). The second one likely carries the wind information, while the directional cues from the visual system (if not optic flow based) are likely mediated via the PB. The resolution of the NO is right versus left, while the resolution of the PB is 360degreees/8.

Showing only four directions of visual input likely misses the peak response in about half the neurons (the ones tuned to the diagonals) in the first set of experiments, if they are driven by the PB. Much of the variability in the visual response might result from this.

Additionally, for PEN cells a single stripe that is flashed might drive them via their optic flow response, producing an unselective flicker response unrelated to their azimuth tuning.

Finally: The response magnitude of the multimodal stimulus could depend on the phase of the two stimuli, i.e. their azimuth difference (e.g. wind from frontal right might drive the cell only when the visual stripe is left off the animal). When always shown in conjunction, this aspect will be missed. This is not unlikely, as the head direction signal phase depends on the experience of the fly and its sensory environment (see recent papers by Fisher et al., and Jayaraman lab), but the tuning of the PFN cells to the wind is static (as it results from one input cell on either hemisphere).

I do not expect new experiments with different stimuli, as the data are very informative as they are, but please rework the text to account for these possibilities and consider them when drawing the conclusions.

Reviewer #3:

Currier et al., performed patch-clamp recordings from columnar neurons in the fly central complex. They discovered that ventral PFNs respond strongly to wind stimuli and rather than forming a map of wind directions, all P-FNs on one side of the brain respond to wind with a peak tuning of ~45° to the right of the fly and vice versa for ventral P-FNs on the other side of the brain. Silencing ventral P-FNs impairs tethered wind-orienting flight behavior while preserving basic sensory and motor capacities. This is a generally informative study with high quality data.

Essential revisions:

1) The R44B10 Gal4 line targets dorsal P-FNs as well as ventral P-FNs based on Janelia's neuronbridge. Because R44B10-Gal4 gives the most obvious phenotypes of all three Gal4's, it would be more conclusive if the authors verified their results with a different Gal4 line that is more selective to ventral P-FNs, or if they performed new analyses to make more clear that the effects they see with the (more selective) split-Gal4 lines are not as subtle as they currently seem.

2) The authors see minimal tuning to wind in P-EN neurons, which are known to be part of the ellipsoid body compass system. Given that similar wind stimuli induced rotations of the ellipsoid body compass system in a parallel recent publication (Okubo et al., 2020), the authors should discuss (at minimum) or perform additional experiments (ideally) to understand why their experiments yielded no coherent tuning to wind in neurons of the ellipsoid-body compass system (i.e., P-ENs), which seems at odds to the other work (which recorded E-PGs, cells that are typically yoked to P-ENs in their tuning).

3) The conclusions in Figure 4F and 4H seemed quite subtle. It would seem that more N would be useful to verify these trends, or the language used should be made more tentative on these preliminary observations.

4) The authors should perform a statistical test for the results of Figure 6H.

5) In Table 1, the authors observe rather depolarized resting potentials alongside a large variance in resting potentials for their recordings. It should be discussed in more detailed why the standard error for resting Vm is so high across recordings. (It's definitely not a typo, and standard deviation, rather than standard error, is being reported here?) A comparison of the resting Vm variance observed here to the variance of recordings in other cell types, outside of the Cx, in the host lab would be useful. With regard to the depolarized resting potentials, was the liquid-liquid junction potential subtracted? Regardless, resting potentials of -18 mV (in an oscillating cell) or -26 mv (in a non-oscillating cell) seem notably positive. More discussion on why the authors believe the recordings were of high quality, given the depolarized resting potentials for some cell classes should be given.

7) Subsection “FB-mediated action selection during basic sensory orienting”: "In contrast, other groups have suggested that 'basic' orienting behavior directly towards or away from a cue – like stripe fixation and up-/downwind orienting – might not require the Cx (Green et al., 2019; Stone et al., 2017; Okubo et al., 2020). Based on the somewhat restricted effects of compass neuron silencing (Giraldo et al., 2018, Green et al., 2019), they suggest that the Cx is only engaged when utilizing landmark orientation as a reference, for either long-distance flight or homing." Most of these studies were careful to argue that basic-orienting behaviors did not require functional E-PGs, not that the entire Cx was unnecessary. The possibility that some neurons in the fan-shaped body would be needed for basic orienting, as argued by the authors, was left open. The text should be altered accordingly.

8) All tuning curves shown have a set of data points that are double plotted (on the ends). The double-plotted data should be noted, with a star or a highlight of some sort.

9) Figure 2E, please define what curves are being correlated to generate the x and y axes values in the legend. Even in the text, where these are discussed, it is written in subsection “Multi-sensory cues are summed in Cx neurons, with some layer-specific integration variability”: "We then took the point-by-point correlation between the concatenated multi-sensory response and the single modality responses, which yielded a pair of correlation coefficients (ρa for airflow and ρs for stripe)". But you have 3 conditions (odor, wind, visual), so the exact definition of "multisensory response" is ambiguous to the reader. Aren't there multiple types of multisensory responses?

[Editors' note: further revisions were suggested prior to acceptance, as described below.]

Thank you for resubmitting your work entitled "Encoding and control of airflow orientation by a set of *Drosophila* fan-shaped body neurons" for further consideration by *eLife*. Your revised article has been evaluated by Ronald Calabrese as the Senior and Reviewing Editor.

The manuscript has been improved but there are some remaining issues that need to be addressed before acceptance, as outlined below:

Reviewer #3:

One limitation of the current paper is that all stimuli used for physiological measurements were presented in open loop. Past work has shown that closed-loop experience with visual input is important for observing a reliable compass signal in the columnar cell types of the fly central complex. This may be one reason that ventral P-FNs, and other cell types recorded herein, do not exhibit robust compass signals. That said, the fact that ventral P-FNs are uniquely sensitive to wind inputs, among the varying columnar cell types tested, is a compelling new result. Future work should aim to link this wind sensitivity, with compass tuning (measured in closed loop) to create increasingly clear hypotheses on the computational roles of ventral P-FNs in navigation.

I still had a few issues that should be addressed.

1) I previously raised the concern that the membrane potentials reported upon in Table 1 were depolarized and had high cross-cell variance (point 5 in the original review). I am satisfied with the authors' response that the depolarized resting potentials are reasonable because (a) the liquid-liquid junction potential was not subtracted (~13 mV for typical *Drosophila* saline solutions) and (b) the expectation that many of these neurons may need to signal bi-directionally. With regard to the high Vm variance, however, I'm still confused. The authors responded by saying that they mistakenly reported the SD rather than the SEM. However, when I divide the variance measure in the previous paper (which is, apparently, the SD) by √N (N was reported in the column immediately to the left) none of the SEM values I thus calculate match the newly reported values. All the reported values are lower, which means that either the SD values were too big in the first submission or the N values were too small. For example, P-F3N2vs had a Vm of -32.9 {plus minus} 14.7 mV, with an N=4 in the original paper. The new version should be -32.9 {plus minus} 7.35mV, but the new paper instead reports -32.9 {plus minus} 2.1mV. This issue needs to be clarified.

2) In the Introduction, the authors write the following:

"Recordings from different columns suggest that ventral P-FN sensory responses are not organized in a "compass" – where all possible stimulus directions are represented as a map. Instead, ventral P-FNs primarily encode airflow arriving from two directions, approximately 45 degrees to the right and left of the midline."

and then in the Discussion, they also write:

“Because ventral P-FNs receive input in the protocerebral bridge (PB), presumably carrying heading information from the compass system (Franconville et al., 2018), they may be well-poised to perform this computation [of orienting upwind toward an odor source].”

Because the authors themselves acknowledge, in quote 2, that ventral P-FNs are likely to receive, and thus express, heading/compass information, I think that they should soften the tone of quote 1--and several other similarly strong statements they make in the manuscript on this issue--where they state that ventral P-FNs primarily encode airflow over compass cues. My view is that if the authors had a closed-loop setup during recordings, where they could clearly elicit and measure compass tuning in central complex neurons, generally speaking, then they would have observed more robust compass tuning in ventral P-FNs as well (conjunctive with their wind responses). It would make the paper better to acknowledge this possibility at the relevant locations. What follows are a bit more details on this point.

I realize that the authors did sometimes observe compass tuning in their system (e.g., the two E-PG recordings now presented in Figure 2—figure supplement 1) and the authors further raise the possibility that for cells that did not express strong tuning, this could be because the tuning peaks of these cells were perfectly misaligned with the four tested stimulus locations (e.g., a tuning peak at 45 degrees vs. stimuli presented at 0 and 90 degrees). However, another explanation for untuned neurons is even more likely, in my view: by presenting stimuli only in open loop, the E-PG compass system in many flies may never have consistently yoked-to or "trusted" the presented stimuli as robust indicators of heading. With the visual condition specifically, it is my understanding that the authors rotated a their arena to a fixed position in the dark and then turned on the lights; this approach is not expected to yield consistent heading signals in E-PGs (or P-FNs) in part because all the visual cues above and below the fly--i.e., the rest of the room and rig--have not rotated when the lights turn on, and thus if the system yokes to those other visual stimuli, rather than the arena, then the system will have visual evidence when the lights turn on that the fly has not rotated. Moreover, even if the system does yoke to the rotating arena, the offset of the E-PG-bump-angle to the arena-angle might change often, within a recording, without closed-loop feedback to reinforce one offset. This issue will tend to generate apparently-poorly-tuned neurons as well, because of a tuning peak that changes across presentations of the same stimulus.

Note that this issue--that during physiological experiments, the compass signals in the fly CX are often ineffectively yoked to stimuli presented in open loop--is not actually a big problem for the main conclusions of the paper; it might even have helped the authors to visualize the ±45 degrees wind inputs in ventral P-FNs (because compass tuning was variable and averaged out). However, when the authors strongly state that these P-FNs do not represent compass cues, seemingly ever, this is likely to be an overstatement given the data provided and basic expectations from the anatomy and physiology in other papers. The authors essentially acknowledge this point in quote 2 above. As such, I ask that the authors adjust the language in the text to acknowledge the possibility of ventral P-FNs being responsive to compass cues in addition to +45 degrees wind stimuli in discussing their results.

3) Continuing on the above point, in subsection “Distinct sensory representations in different CX compartments “, the authors wrote:

"Thus, in the EB, ring neurons appear to carry landmark signals of diverse modalities that collectively anchor the E-PG heading representation, while P-ENs provide angular velocity signals that rotate this representation in the absence of landmark cues (Green et al., 2017; Turner-Evans et al., 2017; Turner-Evans et al., 2020). This model is consistent with our finding that airflow direction cues were not strongly represented in P-ENs, although they are prominent in E-PGs."

In all models, P-ENs and E-PGs, express the same heading bump at all times. This is true when landmarks are driving the E-PG signal around the EB as well as when the P-ENs are performing their integration function in the dark (or when a fly first experiences a new visual environment and has yet to build its visual map). As such, I know of no extant CX model that could explain E-PGs being tuned to heading without P-ENs also being tuned to heading at the same time. P-ENs should show an additional angular velocity modulation that the E-PGs lack, yes, but P-ENs should also have robust heading signals as well. In other words, P-ENs are conjunctively tuned to heading and angular velocity, whereas E-PGs are uniquely tuned to heading. (This is the same point I was making above for P-FNs, which I expect to be conjunctively tuned to heading and {plus minus}45 degrees wind inputs.) As such, I don't find the logic of the above paragraph compelling. If the authors find themselves in a situation where E-PGs, but not P-ENs, are tuned to heading, that is a situation which has no precedence in other papers to date, or models for how the E-PGs and P-ENs work in tandem. I think it is more likely that the compass tuning in their prep, overall, is relatively unstable and variable (in both E-PGs and P-ENs). Again, i think that this instability is fine for the purposes of the conclusions of this paper, but it should just be acknowledged. Please alter the last sentence above, in light of these comments.

4) In subsection “Airflow dominates responses to directional sensory cues in a set of CX columnar neurons”, the authors write:

"In contrast to this strong directional preference in the airflow condition, tuning strength in the visual condition was relatively weak across recorded cell types (Figure 1—figure supplement 2). One notable exception were P-ENs, which displayed modest visual tuning, in agreement with previous results (Green et al., 2017; Fisher et al., 2019)."

I see no unique boost of the tuning index of P-ENs in Figure 1—figure supplement 2 over other cell types. Barring some statistical evidence to support this point, please remove. Also, there are no P-EN recordings in Fisher et al., 2019, that I could find, and I do not think that it should be cited here. Rather, Turner-Evans et al., (2017) should be cited as this is the only paper I know of with P-EN electrophysiology (rather than just imaging).

5) In the Introduction, the authors wrote:

"Another set of EB neurons, known as P-ENs, rotate this heading representation when the fly turns in darkness (Green et al., 2017; Turner-Evans et al., 2017). Despite these robust representations of navigation-relevant variables, genetic disruption of the EB compass network has only indirect effects on navigation."

I am not sure what is meant by "indirect effects"? Perhaps the authors meant "incomplete" effects? If not, please provide more clarification.

6) Subsection “Ventral P-FN airflow responses are organized as orthogonal basis vectors, rather than as a map or compass”: what are the hemibrain names of the vFBNs you're recording from? In the Materials and methods, please describe the process by which you linked up the cells you recorded from in the VT029515 Gal4 line to cells that are likely to be presynaptic to PF2N3s. If the logic here were described, I couldn't find it.

7) In Figure 2B, the cell IDs (#3 and #4) are now reversed relative to the cell Ids in Figure 2E compared to the original version. I believe that the new version is the one with the mistaken assignment.

8) In subsection “The role of ventral P-FNs in natural behavior”, Ferris and Maimonet al., (2018) was referenced in regard to descending neurons that control flight. There were no descending neurons that control flight characterized in that paper, to my knowledge, and this reference should be removed here.

---

## [Author Response]

Summary:Currier et al., performed patch-clamp recordings from columnar neurons in the fly central complex. They discovered that ventral PFNs respond strongly to wind stimuli and rather than forming a map of wind directions, all P-FNs on one side of the brain respond to wind with a peak tuning of ~45degress to the right of the fly and vice versa for ventral P-FNs on the other side of the brain. Silencing ventral P-FNs impairs tethered wind-orienting flight behavior while preserving basic sensory and motor capacities. This is a generally informative study with high quality data important to understanding navigation in flies and potentially in outer systems.

We would like to thank the reviewers for their positive assessment of our manuscript and for their highly constructive feedback. In this revision we have attempted to address all of the major and minor concerns described here, and we feel that the manuscript is much improved thanks to the reviewers' feedback. Despite covid limitations, we were able to obtain some additional data which we have included here. Specifically, (1) we are including new recordings and imaging data suggesting that the most likely source of wind direction information in ventral P-FNs is LALNO(a) neurons (LNa), highlighting the similarity of this airflow circuit to the optic flow circuit identified in Stone et al., 2017. (2) We performed silencing experiments and a small number of recordings in E-PG compass neurons. Confirming previous studies, we observed tuned responses to both visual and airflow stimuli in E-PGs, indicating that our paradigm is sufficient to observe these responses. However, we found no effects of E-PG silencing on airflow orientation, providing some support for the idea that the silencing effects we observed are specific to ventral P-FNs. In addition, we have substantially rewritten the Discussion to reflect several points raised by the reviewers. All new text in the manuscript is noted in blue for ease of reviewing. We look forward to hearing the reviewers’ assessment of our revised manuscript.

Essential revisions:1) The reviewers were generally concerned because their understanding of the connectomics is that the LNs predominantly provide inputs to the PFNs and not predominantly outputs to the LAL. In reviewer consultation "…the LN neurons in question are typically considered inputs to the CX, not outputs, as posited in the authors' final model. As such, … the authors' model [is] … odd. One main reason to believe that LNs provide inputs to the PFNs is because the connectome reveals these cells to receive thousands of input synapses in the LAL and to have thousands of output synapses in the noduli, on PFNs. That said, formally, the LNs in question do have ~30 output chemical synapses in the LAL, and gap junctions are also possible. So… while far from conventional, … one could, in principle, posit an output role, for those neurons…" There was skepticism that PFNs send signals out of the CX via the LN neurons, and if the authors adhere to it in revision they must make every attempt to rationalize this conception, discuss the # of synapse issue, and minimally introduce other, more conventional, models for how control of behavior by these neurons could work, via intermediaries in the fan-shaped body.

We thank the reviewers for pointing this out, and agree that the model presented was not parsimonious. We have made major revisions to our hypotheses regarding the mechanisms by which ventral P-FNs could influence turning behavior. Additionally, we have now examined the activity of LNa neurons as a potential source of tuned airflow signals for ventral P-FNs. We have included this data in a new Figure 5 that also considers an alternative potential source of tuned airflow signals. Based on these data, it appears likely that ventral P-FNs inherit their “basis vector”-like tuning from LNa neurons, as a consequence of the convergence of ventral P-FN neurites in the NO. This can explain both why all P-FNs in one hemisphere share similar airflow tuning, and why they respond only to airflow, and not to odor (a property shared by LNa neurons).

We emphasize that these results are only correlative, and have been careful with our discussion of these data to reflect that fact.

2) The authors patch-clamped cells in more restrictive split-Gal4's for the ventral PFNs and these lines gave the weaker behavioral phenotypes. R44B10 was used to improve the rigor of the behavioral results. R44B10 has labeling outside of the CX, (it's not the worst line; the cells targeted outside of the CX number in the dozens to hundreds - not bad for a generation-1 Gal4 line - not thousands). Rather than perform additional experiments (as stated in the expert reviews) to ensure that the behavioral effects are mediated by ventral PFNs, the authors could perform additional time-series analyses or depictions of their behavioral genetics results with the three Gal4 lines that they used to date, to make clear that the phenotypes in the two split-Gal4 lines are indeed robust. Our concern is that three left-most columns in Figure 5D just don't look that different, and this is a standard quantification. Discussion should state very explicitly the limitations/concerns about using Gal4 lines in the behavioral experiments.

The reviewers issues with the behavioral data are understandable. We have made the following changes in an effort to ameliorate their concerns:

1) We modified plotting of single fly mean orientation vectors to highlight the differences between lines. The silencing phenotypes in split-GAL4 lines are significant by a considerable margin, and we believe the visual presentation of the data clarifies this.

2) We have added data showing that silencing of E-PGs has no effect on airflow orientation behavior. This contrasts with the moderate effects of silencing single P-FN types.

3) We attempted additional silencing experiments using 3 broad driver lines that target multiple classes of ventral P-FN. Unfortunately, none of these lines were viable when crossed to UASKir2.1.

4) We have clarified the limits of our silencing results in the Discussion, including the caveat that our strongest phenotype was observed in a broader line that has some off-target label.

3) All three expert reviewers agree that the findings described in this study are not well integrated with previous literature, specifically in relation to wind sensing within Central Complex circuits, and in relation to processing of information in other compartments. There was also confusion on the rules underlying multisensory interactions in PFNs and the authors should make very clear that they are examining multimodal interactions in a fixed condition.

We have substantially revised our Discussion to better integrate our results with the existing literature. Specifically, we have included a new Figure 8 that illustrates the different pathways by which airflow information enters the EB and FB (based on our data and that of Okubo et al., 2020), and the different forms these representations take. In particular, we highlight the way airflow signals in the LAL may “branch” into discrete representations — a compass-like representation in E-PGs, and a basis vector-like representation in ventral P-FNs.

We believe that this model clarifies some confusing aspects of our data, such as why we observed little airflow tuning in P-ENs when such tuning is prominent in E-PGs.

In addition, we now note in our discussion of multi-sensory integration that our findings pertain only to the stimuli used in this study, which were always presented from the same direction. We also note in the Discussion that we may have missed some multisensory interactions if neurons were tuned to visual stimuli 45 degrees off from one of the axes we examined in our survey.

The detailed expert reviews amplify upon these three basic concerns and will guide the authors in revision.Reviewer #1:In this study, Currier et al., examine the sensitivity to airflow, a visual landmark and olfactory cues across a limited subset of so-called columnar neurons within the Central Complex (Cx) of the fly, a navigational center of the insect brain. They found that a subset of FB columnar cells displays robust responses to directional airflow, and that activity across the population does not follow an abstract representation of heading as was recently reported for another set of columnar neurons in the Cx. Instead, this population seem to report a signal related to a contrast between sides across the body. Finally, they show that genetic-based manipulations of the activity of these cells promote unstable orientations, while not affecting neither motor programs nor the detection of the location of the airflow source. Instead, the authors propose that an improper selection of the motor program underlies the degradation of the stabilization of the oriented response. Overall, I find that these findings are timely, and supported by the experimental evidence largely. However, I find some disconnection between the physiology and the behavior, and between their findings and previous work on airflow processing in Cx columnar cells. In addition, I think that the way the mulmodal interactions are described may be a bit misleading. The specific comments below touch on these points

We thank the reviewer for this positive assessment and for the constructive suggestions.

Link between physiology and behavior:The apparent distinct representation of the population of ventral P-FNs compared to that reported for P-ENs and E-PGS is an important aspect of this study. Specifically, the lack of a systematic map of airflow direction at ventral P-FNs, but rather a contrast base signal. Notably, the authors make no comment about how this representation may change from the allocentric description of airflow location, to one that incorporates egocentric information (for example, heading). I think this is an important point to emphasize the importance of their findings. I would encourage a discussion along this direction especially because they seem to connect the idea that a simple "bases" representation may be adequate for a moment-by-moment action selection. Because these populations of neurons are proposed to be closer to motor programs rather that abstract representations, an egocentric reference frame may be a more appropriate description of population dynamics. If this were the case, a conjunctive interaction is expected with an updated information about heading. In reality, focusing their analysis in the absence of movement in a head-fixed preparation essentially makes the allocentric and egocentric frames apparently overlapping. By I wonder whether by manipulating different signals, such as the visual and the airflow, and making them appear at different locations, may reveal a multimodal interaction that is link to the transformation between abstract onto a more motor reference frame, of whether the authors think this transformation may happen somewhere else, for example, at the level of DNs.Certainly, the behavioral experiments have an internal signal related to the ongoing movements of the fly, that likely is integrated with the sensory signal. The question is whether this integration happens at the ventral layers of the FB or not.

This is an excellent point that we have tried to address somewhat in the revised Discussion and hope to address more fully in a future study. In our revised manuscript we have made clearer the distinction between the map-like representation of airflow direction observed in E-PGs and the basis-vector representation observed in P-FNs. We now note in the Discussion that these two representations may allow for navigational computations to occur in allocentric and egocentric coordinate frames, respectively. We also now include a discussion of possible future experiments studying the interaction of airflow, optic flow, and heading cues that could provide insight into the function of these cells in more natural settings.

Multimodal interactions:If I understood correctly, all stimuli were placed at the same location. I think this may be an important point in the conclusions and interpretations for multisensory interactions. I would encourage to make more explicit the point that the apparent additive effect of multimodal processing is specifically related to the condition that the spatial information of the stimuli is congruent across modalities. It would be interested to see how these interactions change as a function of a dissonant spatial information across modalities.

We have added a note to this section of the manuscript pointing out that summation applies only to the stimuli we used which were always presented from the same direction.

Integration of the work with previous literature:Although the authors describe and acknowledge previous work on airflow processing in Cx, it is a bit disappointing to see a lack of integration of the present findings with those reported before. Specifically, are these airflow signals also dependent on R1? Why are the authors proposing the ventral-FNs are an alternative entrance point of airflow processing? Are they coming from the antennae or bristles all over the body?, etc. Why do they think that E-PGs but not P-ENs are so robust to airflow? What is/are the expected path/s?

We have included two new figures that provide insight into this question. In this revision we show that LAL-NO(a) neurons (LNa) are—like ventral P-FNs— strongly tuned for wind direction with little response to the addition of odor (new Figure 5). Combined with the anatomy and the similarity of ventral P-FN responses in one hemisphere we think it is likely that these neurons represent the source of wind direction information to ventral P-FNs, although we note that silencing experiments will be required to test this directly. Based on these data, we have developed a circuit diagram (new Figure 8) that illustrates the sources of airflow information to both E-PGs and ventral P-FNs. Briefly, we think the most parsimonious model is that both of these originate in the LAL, but enter the central complex through different pathways, R1 neurons for E-PGs, and LAL-NO(a) neurons for ventral P-FNs. This accounts for the weak airflow responses in P-ENs as they are not the source of airflow information to E-PGs.

Reviewer #2:The manuscript by Currier et al., is an impressive and important contribution to understanding the insect central complex. It provides highly relevant data for understanding the role of this brain structure in guiding navigation behavior based on multisensory input. Using *Drosophila* as their model, the authors perform electrophysiological recordings from many identified CX neurons, in particular the so far not characterized PFN neurons. These neurons have been recently proposed to represent the desired heading of the insect (based on data in bees) and the presented data not only corroborate the general hypotheses of how the CX circuitry might guide behavior, but it presents a wealth of detailed data on the role specific types of PFN neurons play in this context in flies. This will drive the development of ideas about how other subtypes of PFN neurons might respond to sensory information, and thereby will aid the generation of a unified idea of how all aspects of the sensory world perceived by the fly can influence action selection via the CX circuits.The technically challenging work is very well carried out, the data is of high quality, controls are in place, and I enjoyed reading the results part a lot. I have one major issue with stimuli, which, however, is mostly relevant for future studies and will only require re-evaluating some conclusions in the present work.There is one major issue in the paper as it is, and this refers to the discussion and the general conclusions about the role the examined neurons play in the overall CX circuitry. As explained in detail below, this appears to be based on a misinterpretation of the hemibrain data. Interestingly, if revised, the altered conclusions (taking into account the real synaptic distributions of the PFN and LN neurons) fit much more nicely with the data available from other species and push our understanding of the CX into a direction of a generally applicable hypothesis of the neural basis of insect navigation.

We thank the reviewer for the strong assessment of our work and appreciate the very helpful suggestions throughout the manuscript.

Essential revisions:1) (sorry if this point is very long) The authors make use of the recently published connectome data, but they appear to have misinterpreted the polarity of their neurons, leading to conclusions that are opposite to the available data in flies, and also at odds with the data from other species. The final circuit hypothesis and the associated conclusions have to therefore be revised to a large extent.Details: Opposite to what the authors state in the Materials and methods and in the Discussion, the lateral neurons of the noduli, the LN neurons (specifically the LNa and LCNpm) do not have outputs in the LAL and (prime) inputs in the noduli. Within the noduli these cells have only outputs (95 and 98%) and in the LAL these cells have only input synapses. It is correct that they also receive inputs from PFNs in the Noduli, but given the lack of outputs outside the Noduli, these must be local feedback connections, not influencing the overall direction of information flow.The LN neurons hence provide input to the PFN cells and are no possible output pathway. The only output that relays information from the PFN cells to other brain regions is via the FB (as there these cells have almost only output synapses). Targets are columnar neurons that project either to the LAL (PFL1) and to the Crepine (FC1 cells, formerly FQ6 cells). This generally is in line with the information flow suggested by Stone et al., (2017), Honkanen et al., (2019) and Franconville et al., (2019). Here, the PFL cells (or related columnar neurons) are the main output pathway from the CX, combining information from the PB (for PFL cells) and the FB to project to the LAL/CRE regions. The PFN cells are suggested to input onto these output neurons, which integrate multiple sources of information (different PFNs, tangential FB cells, delta7 cells in the PB) and could provide a steering signal to the LAL (creating an imbalance between the summed right and left output).The presented data is highly consistent with this idea and the evidence for downstream processing (e.g. DNa02 neurons) is not contradictory to this circuit either (as it receives input from PFL cells).

We agree with this assessment that the LNa neurons are most likely upstream of ventral P-FNs. We have amended our discussion of this point and now present these neurons as likely inputs. Based on our examination of the hemibrain connectome, there appear to be multiple potential output pathways from ventral P-FNs through the FB (through assorted FQ cell types). Therefore, we have removed our discussion of these precise pathways from the manuscript.

2) Stimuli: The tested neurons likely receive input from two sources: the head direction system (via the PB, potentially inhibitory through the delta7 cells) and an input via the noduli (see major point). The second one likely carries the wind information, while the directional cues from the visual system (if not optic flow based) are likely mediated via the PB. The resolution of the NO is right versus left, while the resolution of the PB is 360degreees/8.Showing only four directions of visual input likely misses the peak response in about half the neurons (the ones tuned to the diagonals) in the first set of experiments, if they are driven by the PB. Much of the variability in the visual response might result from this.

This is an excellent point and we have made two notes about this in our revised Discussion. First, we note that we may have missed visual responses that were 45 degrees off from our 4 stimulus directions in the initial survey. Second, we note that future studies examining specifically the integration of heading, airflow, and optic flow signals may provide more insight into integration in ventral P-FNs.

Additionally, for PEN cells a single stripe that is flashed might drive them via their optic flow response, producing an unselective flicker response unrelated to their azimuth tuning.

Yes, this is possible. We did see some tuning for visual stripes in PENs (the strongest of our previous dataset). We have now additionally examined two E-PG responses and observed strong tuning to the visual stimulus, as previously observed.

Finally: The response magnitude of the multimodal stimulus could depend on the phase of the two stimuli, i.e. their azimuth difference (e.g. wind from frontal right might drive the cell only when the visual stripe is left off the animal). When always shown in conjunction, this aspect will be missed. This is not unlikely, as the head direction signal phase depends on the experience of the fly and its sensory environment (see recent papers by Fisher et al., and Jayaraman lab), but the tuning of the PFN cells to the wind is static (as it results from one input cell on either hemisphere).

This is also an excellent point and we have added a note to our discussion of multi-sensory integration, noting that our conclusions only apply to the situation we have examined where cues are presented from the same direction.

I do not expect new experiments with different stimuli, as the data are very informative as they are, but please rework the text to account for these possibilities and consider them when drawing the conclusions.Reviewer #3:Currier et al., performed patch-clamp recordings from columnar neurons in the fly central complex. They discovered that ventral PFNs respond strongly to wind stimuli and rather than forming a map of wind directions, all P-FNs on one side of the brain respond to wind with a peak tuning of ~45 degrees to the right of the fly and vice versa for ventral P-FNs on the other side of the brain. Silencing ventral P-FNs impairs tethered wind-orienting flight behavior while preserving basic sensory and motor capacities. This is a generally informative study with high quality data.

We thank the reviewer for this positive assessment.

Essential revisions:1) The R44B10 Gal4 line targets dorsal P-FNs as well as ventral P-FNs based on Janelia's neuronbridge. Because R44B10-Gal4 gives the most obvious phenotypes of all three Gal4's, it would be more conclusive if the authors verified their results with a different Gal4 line that is more selective to ventral P-FNs, or if they performed new analyses to make more clear that the effects they see with the (more selective) split-Gal4 lines are not as subtle as they currently seem.

We agree that this is a limitation. We tried to perform silencing experiments with three other lines that label both sets of ventral P-FNs (15E12, 67B06, and 20C08) but all were lethal when crossed to UAS-Kir. We were able to perform silencing experiments using a split-GAL4 line that labels E-PGs and show that, in contrast to results using splits that target PF2N3 and PF1N3s, this silencing produced no change in airflow orientation. Thus, we think these results slightly strengthen our finding. However, we acknowledge that the silencing results with 44B10 could arise from off-target expression. We have included this caveat in our revised Discussion.

2) The authors see minimal tuning to wind in P-EN neurons, which are known to be part of the ellipsoid body compass system. Given that similar wind stimuli induced rotations of the ellipsoid body compass system in a parallel recent publication (Okubo et al., 2020), the authors should discuss (at minimum) or perform additional experiments (ideally) to understand why their experiments yielded no coherent tuning to wind in neurons of the ellipsoid-body compass system (i.e., P-ENs), which seems at odds to the other work (which recorded E-PGs, cells that are typically yoked to P-ENs in their tuning).

We performed additional recordings experiments in a few E-PGs and observed tuned wind responses in some of these, as reported by Okubo et al. In that study, the authors found that silencing R1 neurons abolished most of the wind response in E-PG, indicating that wind input to these neurons comes from ring neurons, not from PENs. We have clarified this circuit organization in a new Figure 8. Therefore, we think the reason that we do not observe strong wind tuning in PENs is that these cells mostly provide angular velocity information required to rotate the heading signal, rather than a wind landmark which comes through R1 cells.

3) The conclusions in Figure 4F and 4H seemed quite subtle. It would seem that more N would be useful to verify these trends, or the language used should be made more tentative on these preliminary observations.

Our intention here was only to note if there were some subtle differences in tuning across columns and we wished to report these as observed. We have softened our conclusions regarding these differences but left the observation that “We did notice some subtle differences in tuning that varied with column.”

4) The authors should perform a statistical test for the results of Figure 6H.

We have removed this panel and the language related to it, as the quantity of data in the “upwind” bin was too small to stand up to statistical scrutiny. We thank the reviewer for noticing this issue.

5) In Table 1, the authors observe rather depolarized resting potentials alongside a large variance in resting potentials for their recordings. It should be discussed in more detailed why the standard error for resting Vm is so high across recordings. (It's definitely not a typo, and standard deviation, rather than standard error, is being reported here?) A comparison of the resting Vm variance observed here to the variance of recordings in other cell types, outside of the Cx, in the host lab would be useful. With regard to the depolarized resting potentials, was the liquid-liquid junction potential subtracted? Regardless, resting potentials of -18 mV (in an oscillating cell) or -26 mv (in a non-oscillating cell) seem notably positive. More discussion on why the authors believe the recordings were of high quality, given the depolarized resting potentials for some cell classes should be given.

We thank the reviewer for their attention to detail related to recording quality. This was indeed a typo, and standard deviation was being reported. To maintain consistency throughout the manuscript, we have updated these numbers to accurately reflect the SEM. We believe that the reviewer will find the accurate values much less concerning.

With regards to the relatively depolarized resting potentials, liquid junction potential was not subtracted so these are likely higher than the actual value. In addition, depolarized resting potentials with signaling both through hyperpolarization and depolarization appears to be a common feature of mechanosensory neurons downstream of wind-sensitive JONs, as shown in Chang et al., 2016, Suver et al., 2019, and Okubo et al., 2020.

7) Subsection “FB-mediated action selection during basic sensory orienting”: "In contrast, other groups have suggested that 'basic' orienting behavior directly towards or away from a cue – like stripe fixation and up-/downwind orienting – might not require the Cx (Green et al., 2019; Stone et al., 2017; Okubo et al., 2020). Based on the somewhat restricted effects of compass neuron silencing (Giraldo et al., 2018, Green et al., 2019), they suggest that the Cx is only engaged when utilizing landmark orientation as a reference, for either long-distance flight or homing." Most of these studies were careful to argue that basic-orienting behaviors did not require functional E-PGs, not that the entire Cx was unnecessary. The possibility that some neurons in the fan-shaped body would be needed for basic orienting, as argued by the authors, was left open. The text should be altered accordingly.

We have substantially rewritten this part of the Discussion to make clear that these studies refer specifically to E-PGs.

8) All tuning curves shown have a set of data points that are double plotted (on the ends). The double-plotted data should be noted, with a star or a highlight of some sort.

We made this clarification to the relevant Figures (Figure 1, Figure 3, Figure 4, Figure 1—figure supplement 2) by adding an orange asterisk to -180 degrees on the axes in question and noting the asterisk’s meaning in the corresponding legends.

9) Figure 2E, please define what curves are being correlated to generate the x and y axes values in the legend. Even in the text, where these are discussed, it is written in subsection “Multi-sensory cues are summed in Cx neurons, with some layer-specific integration variability”: "We then took the point-by-point correlation between the concatenated multi-sensory response and the single modality responses, which yielded a pair of correlation coefficients (ρa for airflow and ρs for stripe)". But you have 3 conditions (odor, wind, visual), so the exact definition of "multisensory response" is ambiguous to the reader. Aren't there multiple types of multisensory responses?

We have updated the legend to specifically define what we mean when we say “multisensory” in this context (airflow + stripe together). We also added this clarification to the main text.

[Editors' note: further revisions were suggested prior to acceptance, as described below.]

The manuscript has been improved but there are some remaining issues that need to be addressed before acceptance, as outlined below:

We thank the reviewers for their remaining comments and have addressed each of them as described in detail below.

Reviewer #3:One limitation of the current paper is that all stimuli used for physiological measurements were presented in open loop. Past work has shown that closed-loop experience with visual input is important for observing a reliable compass signal in the columnar cell types of the fly central complex. This may be one reason that ventral P-FNs, and other cell types recorded herein, do not exhibit robust compass signals. That said, the fact that ventral P-FNs are uniquely sensitive to wind inputs, among the varying columnar cell types tested, is a compelling new result. Future work should aim to link this wind sensitivity, with compass tuning (measured in closed loop) to create increasingly clear hypotheses on the computational roles of ventral P-FNs in navigation.

We agree and are pursuing these experiments concurrently.

I still had a few issues that should be addressed.1) I previously raised the concern that the membrane potentials reported upon in Table 1 were depolarized and had high cross-cell variance (point 5 in the original review). I am satisfied with the authors' response that the depolarized resting potentials are reasonable because (a) the liquid-liquid junction potential was not subtracted (~13 mV for typical *Drosophila* saline solutions) and (b) the expectation that many of these neurons may need to signal bi-directionally. With regard to the high Vm variance, however, I'm still confused. The authors responded by saying that they mistakenly reported the SD rather than the SEM. However, when I divide the variance measure in the previous paper (which is, apparently, the SD) by √N (N was reported in the column immediately to the left) none of the SEM values I thus calculate match the newly reported values. All the reported values are lower, which means that either the SD values were too big in the first submission or the N values were too small. For example, P-F3N2vs had a Vm of -32.9 {plus minus} 14.7 mV, with an N=4 in the original paper. The new version should be -32.9 {plus minus} 7.35mV, but the new paper instead reports -32.9 {plus minus} 2.1mV. This issue needs to be clarified.

We want to again thank the reviewer for their attention to detail regarding our recording quality, as it helped us to catch an error in the original version of our analysis script. That error was resolved, and the new values reported represent the true SEM for each recorded cell type. Our MATLAB code for this analysis reads:

> semVm=std(flymeanVm)/sqrt(flycount);

where the flymeanVm variable contains each fly’s mean Vm, and flycount is *N*. Unfortunately, the older computation was directly corrected and is no longer something we can re-check. So we can say for certain that (1) the current values are correct SEMs; and (2) based on your scrutiny of the numbers, the originally reported values do not represent the SD of the now correctly reported SEMs.

From this we can conclude that the original values were not Vm standard deviation across flies. We apologize for casually accepting the reviewer’s suggestion that we may have reported SD in the original reviews — it appears as though the original error was something different. One possibility that seems likely is that we were considering the SD of a single fly’s Vm (over time) in each group, instead of looking at the SD of mean Vm across flies. But we are unfortunately unable to confirm this idea for the reasons outlined above. We hope that this puts the reviewer’s concerns to rest, and again apologize for our poor communication regarding this point.

2) In the Introduction, the authors write the following:"Recordings from different columns suggest that ventral P-FN sensory responses are not organized in a "compass" – where all possible stimulus directions are represented as a map. Instead, ventral P-FNs primarily encode airflow arriving from two directions, approximately 45 degrees to the right and left of the midline."and then in the Discussion, they also write:“Because ventral P-FNs receive input in the protocerebral bridge (PB), presumably carrying heading information from the compass system (Franconville et al., 2018), they may be well-poised to perform this computation [of orienting upwind toward an odor source].”Because the authors themselves acknowledge, in quote 2, that ventral P-FNs are likely to receive, and thus express, heading/compass information, I think that they should soften the tone of quote 1--and several other similarly strong statements they make in the manuscript on this issue--where they state that ventral P-FNs primarily encode airflow over compass cues. My view is that if the authors had a closed-loop setup during recordings, where they could clearly elicit and measure compass tuning in central complex neurons, generally speaking, then they would have observed more robust compass tuning in ventral P-FNs as well (conjunctive with their wind responses). It would make the paper better to acknowledge this possibility at the relevant locations. What follows are a bit more details on this point.I realize that the authors did sometimes observe compass tuning in their system (e.g., the two E-PG recordings now presented in Figure 2—figure supplement 1) and the authors further raise the possibility that for cells that did not express strong tuning, this could be because the tuning peaks of these cells were perfectly misaligned with the four tested stimulus locations (e.g., a tuning peak at 45 degrees vs. stimuli presented at 0 and 90 degres). However, another explanation for untuned neurons is even more likely, in my view: by presenting stimuli only in open loop, the E-PG compass system in many flies may never have consistently yoked-to or "trusted" the presented stimuli as robust indicators of heading. With the visual condition specifically, it is my understanding that the authors rotated a their arena to a fixed position in the dark and then turned on the lights; this approach is not expected to yield consistent heading signals in E-PGs (or P-FNs) in part because all the visual cues above and below the fly--i.e., the rest of the room and rig--have not rotated when the lights turn on, and thus if the system yokes to those other visual stimuli, rather than the arena, then the system will have visual evidence when the lights turn on that the fly has not rotated. Moreover, even if the system does yoke to the rotating arena, the offset of the E-PG-bump-angle to the arena-angle might change often, within a recording, without closed-loop feedback to reinforce one offset. This issue will tend to generate apparently-poorly-tuned neurons as well, because of a tuning peak that changes across presentations of the same stimulus.Note that this issue--that during physiological experiments, the compass signals in the fly CX are often ineffectively yoked to stimuli presented in open loop--is not actually a big problem for the main conclusions of the paper; it might even have helped the authors to visualize the ±45 degrees wind inputs in ventral P-FNs (because compass tuning was variable and averaged out). However, when the authors strongly state that these P-FNs do not represent compass cues, seemingly ever, this is likely to be an overstatement given the data provided and basic expectations from the anatomy and physiology in other papers. The authors essentially acknowledge this point in quote 2 above. As such, I ask that the authors adjust the language in the text to acknowledge the possibility of ventral P-FNs being responsive to compass cues in addition to +45 degrees wind stimuli in discussing their results.

We agree with the reviewer’s main point— that vPFNs might well show heading-related activity if airflow were presented in closed loop. In the present revision we used the terms “compass” and “map” to refer to an organization of sensory information in which all possible airflow directions are represented, as has previously been observed for both visual and wind cues in EPGs. For example, Fisher and Wilson, 2019 measured visual responses in E-PGs in open loop and found that these were organized as a map, with different all possible positions represented (Fig, 1G). Similarly, Okubo et al., 2020 presented open loop wind stimuli in random order while imaging from E-PGs and found the phase of the E-PG bump was linearly related to wind orientation with all possible wind orientations represented by different E-PG phases (Figure 1G). In contrast, we found that all measured PF2N3 neurons in the same hemisphere shared similar open-loop sensory tuning. We did not mean to imply that a heading compass would never be observed in these cells, if they were recorded during closed loop behavior. Indeed, this would be one prediction and is the reason we suggest these experiments at the end of the Discussion.

To clarify the distinction between a sensory map and a heading compass, we have made the following revisions:

1) We added the phrase “in open loop” to the paragraph noted in quote 1.

2) We have revised quote 1 to “Recordings from different columns suggest that ventral P-FN sensory responses are not organized in a sensory “compass” or “map” — where all possible stimulus directions are represented (Fisher et al., 2019, Okubo et al., 2020).

3) We have added a sentence to the Discussion noting that in closed loop stronger heading-related visual signals might be observed in ventral P-FNs.

4) We have clarified in the Discussion that future experiments looking at wind direction and heading should be performed in closed loop.

3) Continuing on the above point, in subsection “Distinct sensory representations in different CX compartments “, the authors wrote:"Thus, in the EB, ring neurons appear to carry landmark signals of diverse modalities that collectively anchor the E-PG heading representation, while P-ENs provide angular velocity signals that rotate this representation in the absence of landmark cues (Green et al., 2017; Turner-Evans et al., 2017; Turner-Evans et al., 2020). This model is consistent with our finding that airflow direction cues were not strongly represented in P-ENs, although they are prominent in E-PGs."In all models, P-ENs and E-PGs, express the same heading bump at all times. This is true when landmarks are driving the E-PG signal around the EB as well as when the P-ENs are performing their integration function in the dark (or when a fly first experiences a new visual environment and has yet to build its visual map). As such, I know of no extant CX model that could explain E-PGs being tuned to heading without P-ENs also being tuned to heading at the same time. P-ENs should show an additional angular velocity modulation that the E-PGs lack, yes, but P-ENs should also have robust heading signals as well. In other words, P-ENs are conjunctively tuned to heading and angular velocity, whereas E-PGs are uniquely tuned to heading. (This is the same point I was making above for P-FNs, which I expect to be conjunctively tuned to heading and {plus minus}45 degrees wind inputs.) As such, I don't find the logic of the above paragraph compelling. If the authors find themselves in a situation where E-PGs, but not P-ENs, are tuned to heading, that is a situation which has no precedence in other papers to date, or models for how the E-PGs and P-ENs work in tandem. I think it is more likely that the compass tuning in their prep, overall, is relatively unstable and variable (in both E-PGs and P-ENs). Again, i think that this instability is fine for the purposes of the conclusions of this paper, but it should just be acknowledged. Please alter the last sentence above, in light of these comments.

Here we are trying to understand how it is that E-PGs show strong wind tuning (in Okubo et al., 2020) but P-ENs do not (in our survey). This was a question raised in the last round of review.

Our working hypothesis is that the open-loop wind responses in E-PGs (shown in Okubo et al., 2020 Figure 1C) arise mainly through R1 neurons, as shown in Okubo et al., 2020 Figure 2F. If R1 neurons deliver wind responses to E-PGs, then P-ENs need not carry strong wind signals. We are not implying that P-ENs do not carry heading signals. That is why we say that these landmark cues “anchor” the heading representation. To clarify this distinction between heading representations (which we do not measure in this paper) and open-loop sensory responses to wind (which both we and Okubo measure) we have changed the sentence noted above to: “This model is consistent with the fact that we did not observe strong sensory responses to airflow in P-ENs, although they are prominent in E-PGs (Okubo et al., 2020).”

4) In subsection “Airflow dominates responses to directional sensory cues in a set of CX columnar neurons”, the authors write:"In contrast to this strong directional preference in the airflow condition, tuning strength in the visual condition was relatively weak across recorded cell types (Figure 1 —figure supplement 2). One notable exception were P-ENs, which displayed modest visual tuning, in agreement with previous results (Green et al., 2017; Fisher et al., 2019)."I see no unique boost of the tuning index of P-ENs in Figure 1—figure supplement 2 over other cell types. Barring some statistical evidence to support this point, please remove. Also, there are no P-EN recordings in Fisher et al., 2019, that I could find, and I do not think that it should be cited here. Rather, Turner-Evans et al., (2017) should be cited as this is the only paper I know of with P-EN electrophysiology (rather than just imaging).

The point we hoped to make here was that visual tuning strength was *weaker* than airflow tuning strength in most cell types, not that it was non-existent. Indeed, as the reviewer points out, many cell types (including P-EN1) do show moderate dynamic range in the visual condition. We added the clause about P-ENs because we didn’t want to give the impression that P-ENs, which are known to have stripe responses, were unresponsive to visual cues in our survey. In order to better communicate these points, we have rephrased the wording so that P-ENs are discussed as an example, rather than as an exception. We have added the Turner-Evans reference here and removed the Fisher reference.

5) In the Introduction, the authors wrote:"Another set of EB neurons, known as P-ENs, rotate this heading representation when the fly turns in darkness (Green et al., 2017; Turner-Evans et al., 2017). Despite these robust representations of navigation-relevant variables, genetic disruption of the EB compass network has only indirect effects on navigation."I am not sure what is meant by "indirect effects"? Perhaps the authors meant "incomplete" effects? If not, please provide more clarification.

We have changed this sentence to “Despite these robust representations of navigation-relevant variables, the EB compass network is not required for all forms of goal-directed navigation.” The next sentence explains what we mean by this: “Silencing E-PGs disrupts menotaxis — straight-line navigation by keeping a visual landmark at an arbitrary angle — but not other kinds of visual orienting (Giraldo et al., 2018; Green et al., 2019).”

6) Subsection “Ventral P-FN airflow responses are organized as orthogonal basis vectors, rather than as a map or compass”: what are the hemibrain names of the vFBNs you're recording from? In the Materials and methods, please describe the process by which you linked up the cells you recorded from in the VT029515 Gal4 line to cells that are likely to be presynaptic to PF2N3s. If the logic here were described, I couldn't find it.

We have re-written this section of the text to clarify our logic:

“We identified two candidate populations that might carry airflow signals to ventral P-FNs.

Using trans-tango experiments, we found that a group of ventral FB neurons (vFBNs) receive input in the antler and appear to be presynaptic to ventral PFNs (Figure 5 —figure supplement 1). The *Drosophila* hemibrain connectome (Scheffer et al., 2020) indicates that P-F_2_N_3_ neurons (PFNa in the hemibrain) receive prominent input from LNa neurons (LAL-NO(a) neurons, Wolff and Rubin, 2018) that receive input in the LAL and project to the third compartment of the NO.”

We have also added our trans-tango data, which was used as motivation for these experiments, as a supplement (Figure 5 —figure supplement 1).

We are not completely sure of the names of vFBNs in the hemibrain connectome dataset, and have therefore not included them here.

7) In Figure 2B, the cell IDs (#3 and #4) are now reversed relative to the cell Ids in Figure 2E compared to the original version. I believe that the new version is the one with the mistaken assignment.

We have corrected this revision error.

8) In subsection “The role of ventral P-FNs in natural behavior”, Ferris and Miamon, (2018) was referenced in regard to descending neurons that control flight. There were no descending neurons that control flight characterized in that paper, to my knowledge, and this reference should be removed here.

We have removed this reference.